



# PIBM 1.0: An individual-based model for simulating phytoplankton acclimation, diversity, and evolution in the ocean

Iria Sala[1] and Bingzhang Chen[1]

[1]Department of Mathematics and Statistics, University of Strathclyde, G1 1XH Glasgow, United Kingdom

**Correspondence:** Bingzhang Chen (bingzhang.chen@strath.ac.uk)

**Abstract.** Phytoplankton is a diverse group of photosynthetic organisms and accounts for almost half of global primary production. However, most existing marine ecosystem models incorporate limited phytoplankton diversity, overlook phytoplankton evolution, and treat phytoplankton as concentrations instead of particles. Here we present an individual-based phytoplankton model that captures three dimensions of phytoplankton traits (size, temperature, and light affinities) and allows phytoplankton cells to mutate in a one-dimensional (1D) water column. Other components of this ecosystem include dissolved inorganic nitrogen, twenty size classes of zooplankton, and detritus, all modelled as Eulerian fields. This hybrid plankton model can reproduce the general seasonal patterns of nutrients, chlorophyll, and primary production in the subtropical ocean. We expect that this model will be a useful tool for studying phytoplankton physiology, diversity and evolution in the ocean.

## 1 Introduction

The ocean carbon cycle plays a pivotal role in affecting how the Earth System responds to climate change (Sigman and Boyle, 2000). Phytoplankton is one of the most critical player in the global ocean carbon cycle, particularly in transporting carbon from the surface to the deep ocean, a process known as the biological carbon pump (Ducklow et al., 2000; Sigman and Boyle, 2000). In addition, phytoplankton constitutes the basis of the marine food web and contributes to almost half of global primary production (Field et al., 1998). Given that numerical models are our primary tool to predict how the Earth System will respond to climate change induced by anthropogenic release of carbon dioxide, the phytoplankton models have to capture the main aspects of phytoplankton physiology and ecology.

There are a number of limitations associated with many existing phytoplankton models. First, the majority of plankton models use the Eulerian framework, while in reality phytoplankton cells are dispersed in the water column by turbulence. While Eulerian models are widely used and accepted, they can induce discrepancies due to Jensen's inequality where the average of the mean may not necessarily equal the mean of the average (Baudry et al., 2018; Christensen et al., 2022).

It is still unclear whether the Eulerian model or fixed-depth incubations over- or underestimate primary production if we assume the Lagrangian phytoplankton model represents the ground truth. For example, Barkmann and Woods (1996) suggested





that the primary production estimates based on incubation bottles fixed at certain depths could overestimate the true primary
production by up to 40%, while the computation by Ross and Geider (2009) suggested only a minor difference.

The above problem becomes even murkier when phytoplankton acclimation is taken into account due to the different time scales of acclimation and mixing (Tomkins et al., 2020). Consider the case that phytoplankton acclimate to varying light conditions by changing intracellular carbon-to-chlorophyll (C:Chl) ratios. If the water column mixing rate is faster than the acclimation rate, phytoplankton cells are essentially acclimating to the average light condition experienced throughout their
life cycle, which may exacerbate the effect of Jensen's inequality. By contrast, if the mixing rate is slower than the acclimation rate, phytoplankton cells constantly adjust their intracellular C:Chl ratios which is more similar to the Eulerian scenario in which phytoplankton cells are fixed at depth. In addition to light acclimation, phytoplankton can adjust their intracellular nutrient quota to acclimate to the external nutrient environment (Morel, 1987). Further considering the stochastic nature of the movements of phytoplankton cells, it is challenging if not impossible to quickly tell the difference between the Eulerian and
Lagrangian model outputs.

What complicates things further is that the acclimation time scales (and other rates such as photosynthesis and nutrient uptake) depend on phytoplankton traits such as cell size (Litchman et al., 2009; Edwards et al., 2012). When comparing small against large cells, acclimation may take place much faster for a small cell than a large cell. Unfortunately, most of the Lagrangian phytoplankton models did not consider phytoplankton traits, which leaves more uncertainty in comparing primary
production estimates between Eulerian and Lagrangian models.

It is worth mentioning that phytoplankton have multiple traits in addition to cell size. The well-known Darwin model developed by Mick Follows and his colleagues (Follows et al., 2007; Barton et al., 2010; Ward et al., 2012; Dutkiewicz et al., 2020) incorporates at least three dimensions of phytoplankton traits: cell size that primarily determines nutrient kinetics, optimal temperature that determines phytoplankton preference to temperature, and optimal light that characterizes phytoplankton pref-
erence for light. To our knowledge, there has not been an individual-based phytoplankton Lagrangian model which incorporates these traits.

Here, we introduce a novel depth-resolved 1D-hybrid model designed to analyse the influence of water column dynamics on phytoplankton growth, productivity and diversity. This hybrid model is built upon the common nutrient-phytoplankton-zooplankton-detritus (NPZD) framework but encloses an individual-based (*Lagrangian*) module, that computes the phyto-
plankton community, coupled with an *Eulerian* module, that calculates the vertical distribution of the remaining tracers (nitrogen concentration, zooplankton and detritus) as continuous concentrations. The Lagrangian module simulates the phytoplanktonic community as super-individuals each representing a cluster of clonal phytoplankton cells that are physiologically identical and share a common history. This type of mixed Eulerian and Lagrangian modelling approach has been used beyond plankton modelling, such as in the field of aerosol-cloud interactions (Grabowski et al., 2019; Dziekan and Zmijewski, 2022).
The model's primary currency is nitrogen, but it also estimates the carbon and chlorophyll content of phytoplankton cells. Phytoplankton physiological rates, which characterise each super-individual, vary with time and depth according to nutrient availability, temperature conditions, and light (Geider et al., 1998; Ross and Geider, 2009), and three master traits: cell size (expressed in terms of the maximal carbon content per cell during its life cycle), optimal temperature, and light affinity (expressed



in terms of the initial slope of the Photosynthesis-Irradiance (P-I) curve. The diversity of phytoplankton super-individuals is

sustained by grazing and mutation. The model structure is shown in Fig. 1.

In the following sections, we first describe our ecological model and the differential equations that govern the growth and selection of phytoplankton. Next, we present the main results of the model and discuss its merits and limitations.

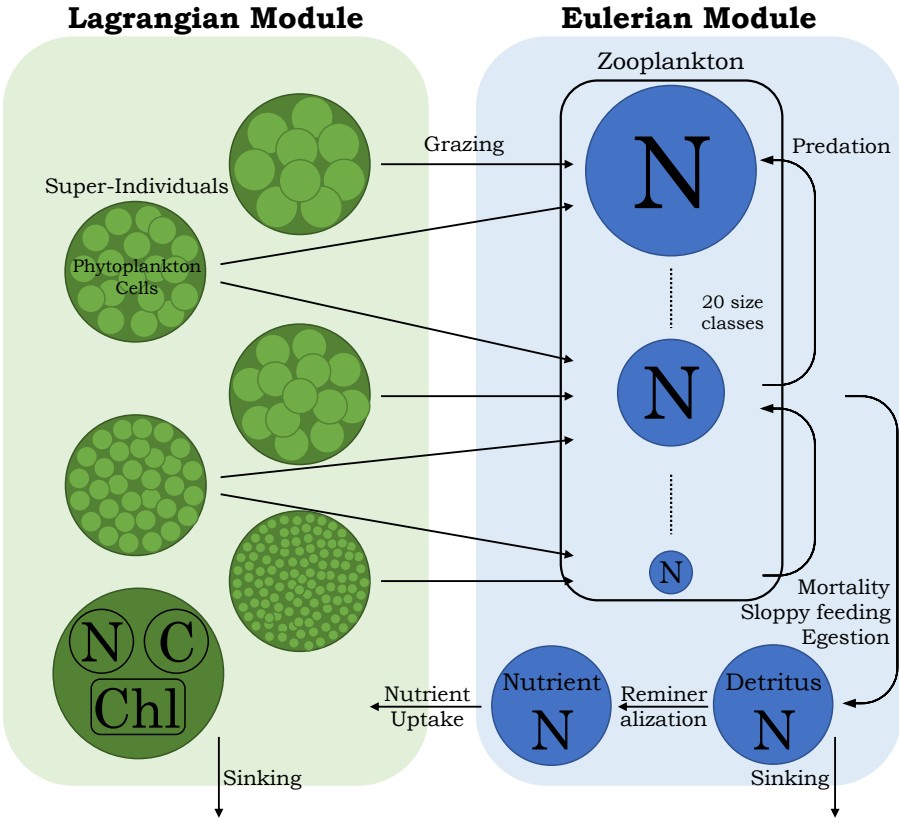

**Figure 1.** Conceptual diagram of the 1D-hybrid Eulerian-Lagrangian model. N: Nitrogen. C: Carbon. Chl: Chlorophyll. The black arrows represent nitrogen flows.

## 2 Model description

### 2.1 Overview

The model is a 1D hybrid model expanded on the classic Nitrogen-Phytoplankton-Zooplankton-Detritus (NPZD) model. Phytoplankton cells are represented by super-individuals (the *Lagrangian* module), which is coupled to an *Eulerian* module, that calculates the dynamics of dissolved inorganic nitrogen (DIN), multiple size classes of zooplankton (ZOO), and detritus (DET)





as continuous concentrations along the vertical domain. In the next subsections, we will describe both modules in detail. All model parameters are listed in Tables 1 and 2.

## 2.2 Lagrangian module

### 2.2.1 Super-Individuals

The *Lagrangian* module simulates the phytoplanktonic community as super-individuals (a cluster of identical phytoplankton cells) to represent a realistic number of phytoplankton cells with affordable computational costs (Scheffer et al., 1995). To avoid memory issues, we assign a constant number of super-individuals throughout the simulation.

### 2.2.2 Phytoplankton model

We assign three master traits, cell size, optimal temperature ($T_{opt}$, $°C$), and the initial slope of the Photosynthesis-Irradiance (P-I) curve ($\alpha^{Chl}$, (W$^2$ m$^{-1}$ g Chl (mol C)$^{-1}$)$^{-1}$ d$^{-1}$) to each phytoplankton super-individual. Cell size determines the capability of phytoplankton to take up inorganic nutrients and its vulnerability to zooplankton grazing. Cell size is expressed in terms of the maximal carbon content per cell ($C_{div}$, pmol C cell$^{-1}$). Note that while the actual cellular carbon content ($P_C$, pmol C cell$^{-1}$) can vary as a result of photosynthesis and respiration depending on its nutrient and light environment, we assume that the traits of nutrient uptake and zooplankton grazing only depend on $C_{div}$ because otherwise these traits will change constantly with time.

The dynamics of phytoplankton cellular carbon ($P_C$), nitrogen ($P_N$, pmol N cell$^{-1}$), and chlorophyll ($P_{Chl}$, pg Chl cell$^{-1}$) are modelled following Geider et al. (1998):

$$\frac{1}{P_C}\frac{dP_C}{dt} = P^C - \zeta\, V_N^C - R_C\, f(T), \tag{1}$$

$$\frac{1}{P_N}\frac{dP_N}{dt} = \frac{V_N^C}{Q^N} - R_N\, f(T), \tag{2}$$

$$\frac{1}{P_{Chl}}\frac{dP_{Chl}}{dt} = \frac{\rho_{Chl}\, V_N^C}{\theta^C} - R_{Chl}\, f(T) \tag{3}$$

$P^C$ (d$^{-1}$) is the carbon-specific photosynthesis rate, $\zeta$ (mol C mol N$^{-1}$) is the cost of biosynthesis, $V_N^C$ (mol N mol C$^{-1}$ d$^{-1}$) is the nitrogen uptake rate, $Q^N$ (mol N mol C$^{-1}$) is the cellular N:C ratio, $\rho_{Chl}$ (dimensionless) is the fraction of phytoplankton carbon production that is devoted to chlorophyll synthesis, $\theta^C$ (g Chl mol C$^{-1}$) is the ratio of Chl synthesis to carbon fixation (representing phytoplankton acclimation to light variability), $R_C$, $R_N$ and $R_{Chl}$ (d$^{-1}$) are the phytoplankton respiration rates for carbon, nitrogen and chlorophyll, respectively. $f(T)$ is the function describing the temperature ($T$, K) dependence of phytoplankton metabolism which is detailed in the temperature section below.





Eq. 1 states that phytoplankton cellular carbon is fueled by photosynthesis that converts inorganic carbon to organic carbon, but is depleted by the cost of both nutrient uptake and respiration. Eq. 2 states that phytoplankton cellular nitrogen is fueled by phytoplankton uptake of nitrogen, but is depleted by respiration. Eq. 3 states that phytoplankton cellular chlorophyll content is fueled by chlorophyll synthesis which depends on both photosynthesis and nitrogen uptake, but is consumed by respiration.

It is important to note that we implicitly assume that $P_C$, $P_N$, and $P_{Chl}$ are not affected by zooplankton grazing which only reduces the number of cells per super-individual.

$P^C$ is a function of light availability as defined below:

$$P^C = P_m^C \left[ 1 - e^{\left( \frac{-\alpha^{Chl}\ \theta^C\ I\ A^\infty}{P_{max}^C} \right)} \right] \tag{4}$$

where $P_m^C$ (d$^{-1}$) is the maximal carbon-specific photosynthesis rate, $I$ (W m$^{-2}$) is the irradiance, and $A^\infty$ (dimensionless) is

the term that accounts for photosynthetic photoinhibition (*see* Eq. 6) which was not included in Geider et al. (1998).

$P_m^C$ depends on intracellular nutrient status:

$$P_m^C = \mu_m \frac{Q^N - Q_{min}^N}{Q_{max}^N - Q_{min}^N} \tag{5}$$

where $\mu_m$ (d$^{-1}$) is the maximal specific growth rate as a function of temperature under resource (nutrient and light) replete conditions (*see* Eq. 10), and $Q_{max}^N$ (mol N mol C$^{-1}$) is the maximal nitrogen-to-carbon ratio and $Q_{min}^N$ is the minimal nitrogen

cell quota. $Q_{max}^N$ and $Q_{min}^N$ are size dependent (Table 2).

To simulate the short-term responses of phytoplankton cells to potential high light stress when being dispersed to the surface, we include photoinhibition into the phytoplankton model following Ross and Geider (2009). Photoinhibition decreases the photosynthesis rate due to the damage of D1 protein under high light, and it is expressed as the fraction of open Photosynthetic Units (PSU) (Han, 2002; Nikolaou et al., 2016):

$$A^\infty = \frac{1}{1 + \sigma_{PSII}\ I\ \tau + K\ \sigma_{PSII}^2\ I^2\ \tau} \tag{6}$$

$\sigma_{PSII}$ (m$^2$ W$^{-1}$) is the effective absorption cross-section of the Photosystem II (PSII) and is parameterized as a power-law relationship with $\theta^C$: $\sigma_{PSII} = \delta\ (\theta^C)^\kappa$ (Nikolaou et al., 2016). $\delta$ ((W m$^{-2}$)$^{-1}$ (g Chl g C$^{-1}$)$^{-1}$) and $\kappa$ (dimensionless) are constants (Table 1). $\tau$ (s) is the turnover time of the electron transfer chain and $K$ (s$^{-1}$) is the ratio of damage to repair constants ($K = k_d/k_r$).

The tension between photo-damage ($k_d$, dimensionless) and repair ($k_r$, s) of a PSU determines the fraction of open reaction centres and abundances of D1 proteins at a given light level. To set up a trade-off between high-light adapted and low-light adapted ecotypes (Moore et al., 1998), we assume that $k_r$ and $\alpha^{Chl}$ are negatively correlated such that phytoplankton cells that are adapted to low-light (i.e., with larger $\alpha^{Chl}$), have a reduced capability of photo-repair (i.e., smaller $K$) (Key et al., 2010). Conversely, phytoplankton cells adapted to high-light (larger $K$) have a greater capability to cope with photoinhibition





but are less efficient in absorbing photons under low light (smaller $\alpha^{Chl}$). In addition, we consider the effect of nutrient status of phytoplankton on cells' ability to photo-repair as nutrient limitation can jeopardize photosynthetic energy transfer efficiency (Herrig and Falkowski, 1989). Therefore, we come up with the following relationship describing $k_r$:

$$k_r = a\left(\frac{\alpha^{Chl}}{b}\right)^v \frac{Q^N - Q^N_{min}}{Q^N_{max} - Q^N_{min}}, \tag{7}$$

in which $a = 2 \times 10^{-5}$, $b = 5 \times 10^{-7}$, and $v = -6.64$ are constants. These parameters are obtained via fitting to the data of *Prochlorococcus* in Moore et al. (1998).

Phytoplankton nitrogen uptake ($V^C_N$) depends on both external levels of DIN and intracellular nitrogen status:

$$V^C_N = V^C_m \frac{\text{DIN}}{\text{DIN} + K_N} \left(\frac{Q^N_{max} - Q^N}{Q^N_{max} - Q^N_{min}}\right)^n \tag{8}$$

with $V^C_m = \mu_m Q^N_{max}$ (mol N mol C$^{-1}$d$^{-1}$) being the maximal specific nitrogen uptake rate, and $K_N$ ($\mu M$) being the half-saturation constant for DIN uptake. $n$ (dimensionless), varying between 0 and 1, is the shape factor adjusting the dependence of the maximum uptake rate ($V^C_m$) on cell quota ($Q^N$) (Geider et al., 1998). All three parameters, $K_N$, $Q^N_{max}$, and $Q^N_{min}$, depend on cell size following allometric relationships (Table 2).

$\rho_{Chl}$ (dimensionless) depends on light, photosynthetic rate, the initial slope of the P-I curve ($\alpha^{Chl}$), and Chl:C ratio ($\theta^C$):

$$\rho_{Chl} = \theta^N_{max} \frac{P^C}{\alpha^{Chl} \theta^C I} \tag{9}$$

where $\theta^N_{max}$ is the maximum chlorophyll to nitrogen ratio (g Chl mol N$^{-1}$). During dark hours when $I = 0$, $\rho_{Chl}$ is assumed to be equal to the value calculated at the end of the preceding light period.

The maximal growth rate, $\mu_m$, depends on $T_{opt}$ (K) as well as environmental temperature ($T$, K) following (Chen, 2022) which extends from the Metabolic Theory of Ecology (Dell et al., 2011; Chen and Laws, 2017):

$$\mu_m = \mu_0 \frac{E_a + E_d}{E_d} \frac{e^{E_a(x-\theta)}}{1 + \frac{E_a}{E_d}e^{(E_a + E_d)(x-\theta)}} \tag{10}$$

in which $\theta$ and $x$ are the transformed optimal and environmental temperatures (for mathematical convenience), respectively ($\theta = \frac{1}{k_b}(\frac{1}{T_0} - \frac{1}{T_{opt}})$ and $x = \frac{1}{k_b}(\frac{1}{T_0} - \frac{1}{T})$, with $T_0$ (K) being the reference temperature at 288 K). $\mu_0$ (d$^{-1}$) is the normalization constant for $\mu_m$ ($\theta = 0$), $E_a$ (eV) is the intraspecific activation energy, and $E_d$ (eV) is the nominal activation energy regulating how fast the growth rate ($\mu$, d$^{-1}$) decreases with $T$ when $x > \theta$. $k_b$ (eV K$^{-1}$) is the Boltzmann constant.

Based on analyzing a dataset of phytoplankton growth against temperature, $\mu_0$ (d$^{-1}$), $E_a$ (eV), and $E_h$ (eV) are found to be allometric functions of $\theta$ (Chen, 2022):

$$\mu_0 = \mu' \, e^{E_i\theta} \tag{11a}$$

$$E_a = E_{a0} \, e^{\beta\theta} \tag{11b}$$

$$E_h = E_{h0} \, e^{\phi\theta} \tag{11c}$$





in which $\mu'$ (d$^{-1}$) is the normalization constant for $\mu_0$ when $\theta = 0$, $E_i$ (eV) is the interspecific activation energy, $E_{a0}$ (eV) is the normalization constant for $E_a$ when $\theta = 0$, $\beta$ (eV) is the scaling exponent against $\theta$ for $E_a$, $E_{h0}$ (eV) is the normalization constant for $E_h$ when $\theta = 0$, and $\phi$ (eV) is the scaling exponent for $E_h$.

Following Wirtz (2011), we assume that $\mu'$ varies with cell Equivalent Spherical Diameter (ESD, $\mu$m) as a result of intracellular self-shading and excess density:

$$\mu' = \frac{\mu'_0}{1 + a_0 \left(\frac{\rho^s}{\rho_0}\right)^{\frac{1}{3}} \text{ESD}} \tag{12}$$

where $a_0$ ($= 0.34$ m$^{-1}$) is the length scale of photosynthesis depletion, $\rho^s$ ($= 0.25$ pg C m$^{-3}$) is the carbon density at a reference ESD ($ESD_s = 8.00$ $\mu$m), $\rho_0$ ($= 0.50$ pg C m$^{-3}$) is the specific carbon density for the relative chloroplast volume ($V_{Chl}/V$).

We also assume that phytoplankton respiration ($R_C$, $R_N$, and $R_{Chl}$) follow the same temperature dependence as $\mu_m$ (Barton et al., 2020).

Following Wirtz (2011), we assume that these specific respiration rates scale inversely with phytoplankton ESD:

$$R_C = R_{C,s} \frac{\text{ESD}_s}{\text{ESD}} \tag{13}$$

where $R_{C,s}$ is the temperature dependent respiration rate at ESD$_s$. If $T = T_{opt}$, $R_{C,s} = 0.025$ d$^{-1}$ for the specific respiration rates of phytoplankton carbon, nitrogen and chlorophyll.

Eqs. 12 and 13 capture the unimodal relationship between maximal growth rate and cell size (Chen and Liu, 2010, 2011; Marañón et al., 2013). If a phytoplankton cell is too large, its self-shading and intracellular decline of CO$_2$ reduces its maximal growth rate. On the other hand, if a cell is too small, the high specific respiration cost leads to a rapid decline in maximal growth rate. This constraint on the range of cell size plays a fundamental role in shaping phytoplankton size structure in the model in which the phytoplankton cells are allowed to evolve freely.

### 2.2.3 Phytoplankton division, death, and evolution

Phytoplankton cell division takes place when the cellular carbon content reaches $C_{div}$ (Cianelli et al., 2009; Ross and Geider, 2009). When a cell divides, the parent cell is split into two equal daughter cells, each inheriting half of the carbon, nitrogen, and chlorophyll content. As a consequence, the number of cells per super-individual doubles.

Phytoplankton cell dies when phytoplankton cellular carbon falls below a minimal threshold ($C_{min}$, pmol C cell$^{-1}$) defined as a quarter of $C_{div}$ (Ross and Geider, 2009) or when the total amount of nitrogen of a super-individual drops below 0.10% of the average nitrogen content of all super-individuals.

It is important to note that in addition to respiratory cost, phytoplankton cells suffer from zooplankton grazing. We assume that zooplankton grazing can only reduce the number of cells per super-individual without affecting cellular carbon or nitrogen.

When a cell dies, its nitrogen content is converted into detritus. At the same time, to keep the number of super-individuals constant, the super-individual with the maximum nitrogen content is divided into two new super-individuals each with half of the number of cells of the parent super-individual.





When phytoplankton cells divide, they are allowed to have a small probability to mutate (i.e., changing the values of the three

traits $log(C_{div})$, $T_{opt}$, and $log(\alpha^{Chl})$) following a Gaussian distribution with the mean equal to the parent cell's trait value and a given standard deviation. The probability of a super-individual to mutate is proportional to its number of cells. Although it may not accurately reflect reality, for the sake of simplicity and ease of modelling, we assume that all ecotypes share the same mutation rate. However, in reality, mutation rates can vary among different phytoplankton ecotypes (Beardmore et al., 2011), and even within the same species when subjected to stress (Bjedov et al., 2003). Besides, while we assume that the mutation of

one trait is independent of others, the user can modify the mutation covariance matrix to change how the mutation of one trait can depend on those of other traits.

### 2.2.4 Mean trait and trait (co)variance

We characterize phytoplankton community composition in terms of mean trait and trait covariance. Trait covariance can be used to represent part of functional diversity (Norberg et al., 2001; Chen et al., 2019).

The community mean phytoplankton trait is calculated as the carbon-weighted mean of all phytoplankton cells in the community:

$$\bar{l} = \frac{\sum_{i=1}^{k} l_i \, n_i \, P_{C,i}}{\sum_{i=1}^{k} n_i \, P_{C,i}} \tag{14}$$

in which $l_i$ and $n_i$ represent the trait value and the number of cells of super-individual $i$, respectively. $P_{C,i}$ represents the cellular carbon content of super-individual $i$.

The trait covariance is calculated as:

$$\text{COV}(l_j, l_m) = \frac{\sum_{i=1}^{k} (l_j - \overline{l_j}) \, (l_m - \overline{l_m}) \, n_i \, P_{C,i}}{\sum_{i=1}^{k} n_i \, P_{C,i}} \tag{15}$$

where $\overline{l_j}$ and $\overline{l_m}$ represent the mean trait value of trait $j$ and $m$, respectively. It is important to note that we treat $log(C_{div})$, $T_{opt}$, and $log(\alpha^{Chl})$ as traits as they are more likely to follow normal distribution than the raw units.

### 2.2.5 Functional diversity

We calculated the functional richness and evenness of phytoplankton cells using the R package **TPD** which considers intraspecific trait variability by computing multidimensional trait probability densities using Gaussian kernels (Mason et al., 2005; Carmona et al., 2019).

In brief, the functional richness is calculated as the amount of functional space occupied by all phytoplankton superindividuals in a community, similar to the volume calculated by the hypervolume approach (Blonder et al., 2018). The functional

evenness measures how even the relative abundances of different trait values are in a community.

### 2.3 Eulerian module

Dynamics of dissolved inorganic nitrogen, zooplankton, and detritus are modelled as Eulerian fields.





### 2.3.1 Dissolved inorganic nitrogen

The temporal and spatial variability of dissolved inorganic nitrogen (DIN, mmol N m$^{-3}$) depends on the total nutrient uptake
by phytoplankton ($P_{\text{uptake}}$, mmol N m$^{-3}$; *see* Eq. 17), zooplankton excretion ($Z_{\text{exc}}$; *see* Eq. 18) and detritus regeneration
($D_{\text{reg}}$; *see* Eq. 19), and vertical diffusion (last term of Eq. 16):

$$\frac{\partial \text{DIN}}{\partial t} = -P_{\text{uptake}} + Z_{\text{exc}} + D_{\text{reg}} + \frac{\partial}{\partial z}\left[K_v(z)\frac{\partial \text{DIN}}{\partial z}\right] \tag{16}$$

where $K_v(z)$ (m$^2$ s$^{-1}$) is the vertical eddy diffusivity at each depth layer. $P_{\text{uptake}}$ at grid $s$ during the time step $\Delta t$ is the sum
of the nutrients taken up by all super-individuals within grid $s$:

$$P_{\text{uptake}} = \frac{1}{H(s)} \sum_{i=1}^{k}\left[\left(P_{N,i}(t+\Delta t) - P_{N,i}(t)\right)n_i(t+\Delta t)\right] \tag{17}$$

in which $H(s)$ (m) is the height of the vertical grid $s$, $P_{N,i}(t+\Delta t)$ and $P_{N,i}(t)$ (mmol N per cell) represent the cellular nitrogen
content of super-individual $i$ at time $t+\Delta t$ and $t$, respectively, and $n_i(t+\Delta t)$ represents the number of cells associated with
the super-individual $i$ at time $t+\Delta t$.

We assume that $Z_{\text{exc}}$ is a constant fraction of the total amount of food ingested by all zooplankton:

$$Z_{\text{exc}} = (1-\xi-\eta)\sum_{j=1}^{N_z} I_j \, \text{ZOO}_j \tag{18}$$

where $\xi$ (dimensionless) is the gross growth efficiency of zooplankton, assumed constant for each size class. $\eta$ (dimensionless)
is the fraction of unassimilated food by zooplankton, also assumed constant for each size class. $I_j$ ($d^{-1}$) is the *per capita* total
ingestion rate of zooplankton size class $j$ and is elaborated in the following section.

$D_{\text{reg}}$ is a linear function of detritus concentration (DET) as well as temperature:

$$D_{\text{reg}} = R_{dn} \, \text{DET} \, f_h(T) \tag{19}$$

where $R_{dn}$ (d$^{-1}$) is the conversion rate from detritus to dissolved inorganic nitrogen, and $f_h(T)$ describes the temperature
dependence of heterotrophic activities including zooplankton grazing and detritus regeneration formulated according to the
Arrhenius equation:

$$f_h(T) = e^{\frac{E_z}{k_b}\left(\frac{1}{T_{ref}} - \frac{1}{T}\right)} \tag{20}$$

where $E_z$ (eV) is the activation energy for heterotrophic processes (Table 1).

### 2.3.2 Zooplankton

The model resolves 20 size classes of zooplankton spaced uniformly in log space from 0.80 to 3600 $\mu$m ESD. We define the
smallest size class as 0.80 $\mu$m to mimic the smallest heterotrophic eukaryotes in the ocean that predominantly feed on bacteria.
The upper limit of 3600 $\mu$m is selected as a result of a tradeoff between providing appropriate grazers that can feed on large





phytoplankton and computing demand as we wish to fix the number of zooplankton size classes as 20. Because phytoplankton cells are allowed to freely evolve in the simulation, it is possible that some phytoplankton cells can evolve into an extremely large (or small) size that no zooplankton can feed on. But we cannot design a size range of zooplankton too large because otherwise there would be too few zooplankton size classes within the realistic zooplankton size range.

The nitrogen biomass of each zooplankton size class ($\mathrm{ZOO}_j$, mmol N m$^{-3}$) increases with the amount of prey (including all

phytoplankton super-individuals and zooplankton smaller than themselves) they can eat, but is reduced by the predation from larger zooplankton in additional to the natural mortality.

$$\frac{\partial \mathrm{ZOO}_j}{\partial t} = \mathrm{ZOO}_j\, \xi_j \sum_{j_{prey}=1}^{J} I_{j,j_{prey}} - \sum_{j_{pred}=1+j}^{N_z} \mathrm{ZOO}_{j_{pred}} I_{j_{pred},j} - \mathrm{ZOO}_j\, m_z\, f_h(T) + \frac{\partial}{\partial z}\Big(K_v(z)\, \frac{\partial \mathrm{ZOO}_j}{\partial z}\Big) \tag{21}$$

where $m_z$ (d$^{-1}$) is the linear zooplankton mortality rate. $J$ is the total number of prey items including all phytoplankton super-individuals within the grid and other smaller zooplankton.

Zooplankton *per capita* ingestion rate ($I_{j_{pred},j_{prey}}$, d$^{-1}$) is calculated following a sigmoidal functional response, depending on total prey biomass ($B_{j_{prey}}$, mmol N m$^{-3}$) (Ward et al., 2012):

$$I_{j_{pred},j_{prey}} = f_h(T)\, I_{j_{pred}}^{max}\, \frac{\phi_{j_{pred},j_{prey}} B_{j_{prey}}}{F_{j_{pred}} + K_{P,j_{pred}}}\, (1 - e^{\Lambda F_{j_{pred}}}) \tag{22}$$

where $I_{j_{pred}}^{max}$ (d$^{-1}$) is the size-dependent maximum ingestion rate (Table 2) (Hansen et al., 1997; Ward et al., 2012). $\phi_{j_{\mathrm{pred}},j_{\mathrm{prey}}}$ (dimensionless) is the palatability of prey $j_{\mathrm{prey}}$ for predator $j_{\mathrm{pred}}$. $F_{j_{\mathrm{pred}}}$ (mmol N m$^{-3}$) is the total prey availability for predator

$j_{\mathrm{pred}}$, and $K_{P,j_{\mathrm{pred}}}$ (mmol N m$^{-3}$) is the grazing half-saturation constant of predator $j_{\mathrm{pred}}$. The term $(1 - e^{\Lambda F})$ represents the effect of prey refuge which reduces the grazing effort as available prey becomes scarce (Ivlev, 1955; Mayzaud and Poulet, 1978). The total ingestion rate of zooplankton size class $j$ is therefore $I_j = \sum_{j_{prey}=1}^{J} I_{j,j_{prey}}$.

The food availability for the zooplankton size class $j_{\mathrm{zoo}}$ ($F_{j_{\mathrm{zoo}}}$) includes both phyto- and zooplankton prey and is computed as:

$$F_{j_{\mathrm{zoo}}} = \sum_{i_{\mathrm{phy}}=1}^{k} \phi_{j_{\mathrm{zoo}},i_{\mathrm{phy}}}\, B_{i_{\mathrm{phy}}} + \sum_{i_{\mathrm{zoo}}=1}^{j_{\mathrm{zoo}}-1} \phi_{j_{\mathrm{zoo}},i_{\mathrm{zoo}}}\, B_{i_{\mathrm{zoo}}} \tag{23}$$

in which $k$ is the number of super-individuals within the vertical grid. $B_{i_{\mathrm{phy}}} = n_{i_{\mathrm{Phy}}} P_{N,i_{\mathrm{Phy}}}/H$ and $B_{i_{\mathrm{zoo}}}$ (mmol N m$^{-3}$) are the nitrogen biomass of the $i_{\mathrm{Phy}}^{th}$ phytoplankton super-individual and the $i_{\mathrm{zoo}}^{th}$ zooplankton size class, respectively. Note that there is no zooplankton prey for the smallest zooplankton size class ($j_{\mathrm{zoo}} = 1$). We assume that zooplankton do not feed on other zooplankton larger than their own size, but can feed on any phytoplankton prey, although feeding on a phytoplankton

prey larger than their optimal prey size is penalized by the low prey palatability, $\phi_{j_{\mathrm{pred}},j_{\mathrm{prey}}}$ (dimensionless):

$$\phi_{j_{\mathrm{pred}},j_{\mathrm{prey}}} = exp\left[ -\left( \ln\left(\frac{\vartheta_{j_{\mathrm{pred}},j_{\mathrm{prey}}}}{\vartheta_{\mathrm{opt}}}\right)\right)^2 \left(2\sigma_{j_{\mathrm{pred}}}^2\right)^{-1} \right] \tag{24}$$

where $\vartheta_{j_{\mathrm{pred}},j_{\mathrm{prey}}}$ (dimensionless) is the predator:prey volume ratio, $\vartheta_{\mathrm{opt}}$ (dimensionless) is the optimal predator:prey volume ratio (Kiørboe, 2009), and $\sigma_{j_{\mathrm{pred}}}$ (dimensionless) is the standard deviation of the feeding kernel.





$\vartheta_{\mathrm{opt}}$ is estimated from the optimal prey size defined as ESD ($\mathrm{ESD}^{\mathrm{pred}}_{\mathrm{prey}_{\mathrm{opt}}}$) vs. predator ESD ($\mathrm{ESD}_{\mathrm{pred}}$) (Hansen et al., 1994; 270 Banas, 2011):

$$\mathrm{ESD}^{\mathrm{pred}}_{\mathrm{prey}_{\mathrm{opt}}} = 0.65 \ \mathrm{ESD}^{0.56}_{\mathrm{pred}} \tag{25}$$

As we assume zooplankton grazing only affects the number of cells per super-individual ($n_{i_{\mathrm{phy}}}$) instead of cellular carbon or nitrogen, it turns out that it is not a trivial task to estimate the changes in $n_{i_{\mathrm{phy}}}$ to conserve total nitrogen. We assume that the phytoplankton mortality due to zooplankton grazing ($g_{\mathrm{phy}}$, d$^{-1}$), or the proportional loss of nitrogen content, during the time 275 step $\Delta t$, is equal for all phytoplankton cells within the same super-individual (and the same for all phytoplankton cells within the same vertical layer). While the deaths of phytoplankton cells can be a binomial process, we can assume that the loss of cell numbers within a super-individual is proportional to the grazing rate $g_{phy}$ thanks to the law of large numbers (Beckmann et al., 2019). Therefore, we have:

$$n_{i_{\mathrm{phy}}}(t + \Delta t) = n_{i_{\mathrm{phy}}}(t) \left(1 - g_{\mathrm{phy}} \ \Delta t\right) \tag{26}$$

The total phytoplankton biomass loss due to zooplankton grazing within each vertical layer ($P_G$, mmol N m$^{-3}$) can be calculated as (note that we drop the subscript of $z$ for convenience):

$$
\begin{aligned}
P_G &= -\frac{1}{H} \sum_{i_{\mathrm{phy}}=1}^{k} P_{N,i_{\mathrm{phy}}}(t) \left( n_{i_{\mathrm{phy}}}(t) - n_{i_{\mathrm{phy}}}(t + \Delta t) \right) \\
&= -\frac{1}{H} \sum_{i_{\mathrm{phy}}=1}^{k} P_{N,i_{\mathrm{phy}}}(t) \left( n_{i_{\mathrm{phy}}}(t) - n_{i_{\mathrm{phy}}}(t)(1 - g_{\mathrm{phy}} \ \Delta t) \right) \\
&= -\frac{1}{H} \sum_{i_{\mathrm{phy}}=1}^{k} P_{N,i_{\mathrm{phy}}}(t) \ n_{i_{\mathrm{phy}}}(t) \ g_{\mathrm{phy}} \ \Delta t \\
&= g_{\mathrm{phy}} \ \Delta t \ \frac{\sum_{i_{\mathrm{phy}}=1}^{k} P_{N,i_{\mathrm{phy}}}(t) \ n_{i_{\mathrm{phy}}}(t)}{H} \\
&= \sum_{j_{\mathrm{zoo}}=1}^{N_Z} I_{j_{\mathrm{zoo}},i_{\mathrm{phy}}} \ \Delta t
\end{aligned}
\tag{27}
$$

where $k$ is the number of phytoplankton super-individuals within the vertical layer.

Since $\frac{\sum_{i_{\mathrm{phy}}=1}^{k} P_{N,i_{\mathrm{phy}}}(t) \ n_{i_{\mathrm{phy}}}(t)}{H}$ is the total phytoplankton nitrogen concentration within each vertical layer ($P_N$), we can 285 derive:

$$g_{\mathrm{phy}} = \frac{\sum_{j_{\mathrm{zoo}}=1}^{N_Z} I_{j_{\mathrm{zoo}},i_{\mathrm{phy}}}}{P_N} \tag{28}$$

Hence, the number of cells within a super-individual at the next time step ($t + \Delta t$) can be computed as:

$$n_{i_{\mathrm{phy}}}(t + \Delta t) = n_{i_{\mathrm{phy}}}(t) \left(1 - \frac{\sum_{j_{\mathrm{zoo}}=1}^{N_Z} I_{j_{\mathrm{zoo}},i_{\mathrm{phy}}}}{P_N} \ \Delta t \right) \tag{29}$$





It is therefore important that the grazing effect ($n_{i_{phy}}(t + \Delta t)$) has to be computed before phytoplankton nutrient uptake
($P_P$, Eq. 17).

### 2.3.3  Detritus

Changes in the concentration of detritus (DET, mmol N m$^{-3}$) are computed as:

$$\frac{\partial \text{DET}}{\partial t} = \sum_{j=1}^{N_z} \Big( \eta \, I_j + m_z \, f_h(T) \Big) \text{ZOO}_j - R_{dn} \, \text{DET} \, f_h(T) - W_d \, \frac{\partial \text{DET}}{\partial z} + \frac{\partial}{\partial z} \Big( K_v(z) \, \frac{\partial \text{DET}}{\partial z} \Big) \tag{30}$$

where $W_d$ (m d$^{-1}$) is the detritus sinking rate.

### 2.3.4  Phytoplankton biomass, trait distribution, and primary production

The Eulerian module additionally calculates the total phytoplankton nitrogen, carbon and chlorophyll concentration, and the
net primary productivity at each grid layer. The concentration of total phytoplankton nitrogen, carbon and chlorophyll biomass
($B_*; B_{P_N}$, mmol N m$^{-3}$; $B_{P_C}$ mmol C m$^{-3}$; and $B_{P_{CHL}}$ mg Chl m$^{-3}$) is computed at each grid layer as (note that we drop
the subscript of $z$ for convenience):

$$B_* = \frac{\sum_{i=1}^k P^*_{i_{\text{phy}}} \cdot n_{i_{\text{phy}}}}{H}, \tag{31}$$

where $P*_{i_{\text{phy}}}$ refers to the three different units of mass (i.e., $P_N$, $P_C$ and $P_{CHL}$).

Net Primary Productivity ($NPP$, mg C m$^{-3}$ d$^{-1}$) is integrated over a daily cycle:

$$NPP = \frac{1}{H} \sum_{t=0}^{24} \sum_{i=1}^{k(t)} \left[ \Delta P_{C,i}(t) \, n_i(t) \right] \tag{32}$$

where $k(t)$ is the number of phytoplankton super-individuals at time $t$.

### 2.4  Size Spectra

Size spectra have been widely used to provide useful insights into the size structure and energy flow of aquatic communities
(Platt and Denman, 1977). To illustrate how this model can be used to simulate the plankton size distribution, we plotted the
size spectra for both phyto- and zooplankton at surface water and compared them between winter (31$^{st}$ March) and summer
(31$^{st}$ August).

To compute the size spectra, we counted the abundances of phytoplankton and zooplankton into the 20 size bins designed
for zooplankton, although the phytoplankton cells were missing in large size classes.

Zooplankton abundances were estimated as the ratio between the zooplankton carbon divided by the individual carbon
content of each size class. The individual carbon content of zooplankton was estimated from their biovolume following the





allometric relationship proposed by Harris et al. (2000). First, based on the ESD of each zooplankton size class, their cell
volume was calculated:

$$Z_C = \rho_Z \, \frac{\pi}{6} \, \text{ESD}^3 \, p_d \, p_c, \tag{33}$$

where $\rho_Z$ is the density of zooplankton organisms, assumed to be equal to seawater density (1.025 g cm$^{-3}$). $p_d$ (dimensionless)
is the proportion of wet mass constituted by dry mass, and $p_c$ (dimensionless) is the proportion of dry mass constituted by
carbon. We used the $p_d$ and $p_c$ values defined for non-gelatinous zooplankton (0.20 and 0.45, respectively). The carbon content
of each zooplankton size class was converted from zooplankton nitrogen using the Redfield ratio (C:N = 106:16).

The size spectra of both phytoplankton and zooplankton were calculated as Ordinary Least-Squares (OLS) regression lines
with log10(abundance) as the response variable and log10(biovolume) as the predictor. All zero abundance data were removed.
In addition, the abundance data of the two smallest size classes (0.8 $\mu$m, 1.2 $\mu$m) were removed for zooplankton because such
small size classes, deviating from the normal linear trend, were not often considered in the construction of size spectra.

## 2.5 Physical forcing

As a case study, our 1D model simulates the upper 250 m of the Bermuda Atlantic Time-series Study (BATS) station, although
some details (e.g., phosphorus limitation instead of nitrogen limitation) are not considered to keep its generality. A total of 100
depth levels define the vertical grid, with increasing resolution towards the sea surface following a sigma grid, similar to that
used in the Regional Ocean Modeling System (ROMS; Shchepetkin and McWilliams (2005)).

The model is forced with three external environmental variables: temperature, vertical eddy diffusivity ($K_v$), and photosyn-
thetically active radiation (PAR). Temperature profiles are imported from World Ocean Atlas 2013 (Locarnini et al., 2024).
Vertical eddy diffusivity profiles ($K_v$, m$^2$ s$^{-1}$) were interpolated from a previous model output (Bruggeman and Bolding,
2014; Le Gland et al., 2021). We define the Mixed Depth Layer (MLD, m) as the depth at which $K_v < 10^{-4}$ m$^2$ s$^{-1}$.

Surface PAR ($I_0$, W m$^{-2}$) at BATS station was estimated as a function of mid-day light, time of the day, and day length
following Anderson et al. (2015). Light values along the water column ($I_z$, W m$^{-2}$) are estimated at each time step following
the Beer-Lambert law, based on $I_0$ and chlorophyll concentration:

$$I_z = I_0 \, e^{-z \, (K_w + K_{\text{Chl}} \, \int_z^0 \text{Chl}(z) \, dz)}, \tag{34}$$

where $K_w$ (m$^{-1}$) and $K_{\text{Chl}}$ ((mg Chl m$^2$)$^{-1}$) are the attenuation coefficients for seawater and chlorophyll $a$, respectively.

Fig. 2 shows the temporal and spatial distribution of the physical forcing variables, representing the typical seasonal vari-
ability at the BATS station. During winter and early spring, the temperatures were low while $K_v$ were high. The mixed layer
reaches a maximum depth of around 250 m from February to the end of March. Starting from April, an abrupt decrease in
MLD is observed, associated with a decrease in $K_v$ and an increase in temperature in the first 25 m of the water column. From
late spring to late summer, the water column is strongly stratified. It can be observed how the surface mixed layer gets warmer
and reaches a depth $\sim$ 50 m in September. At the beginning of fall, a decrease in temperature and an increase in $K_v$ deepens
the surface mixed layer.





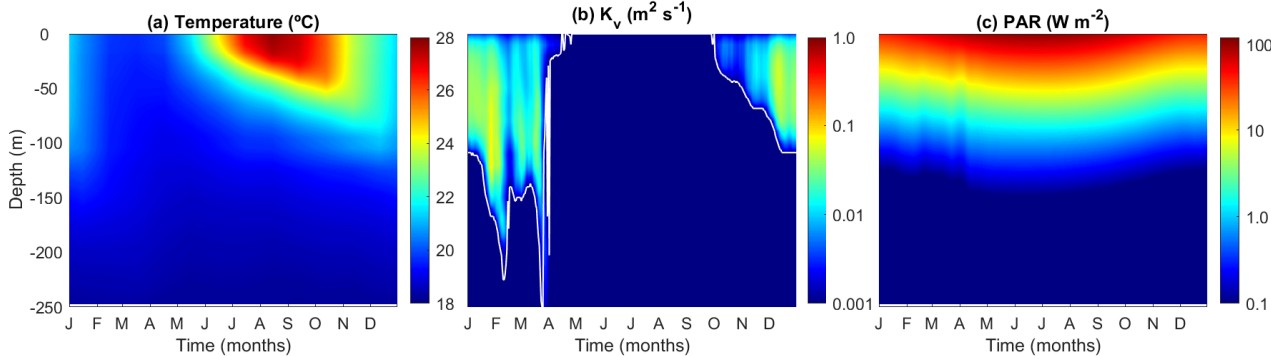

**Figure 2.** Temporal and vertical variability of the model forcing variables. (a) Temperature (°C), (b) Vertical diffusivity ($K_v$, m² s⁻¹), and (c) Photosynthetically active radiation (PAR; W m⁻²). In (b), the white line identifies the mixed layer depth (m).

## 2.6 Vertical movements of particles

The movement of both super-individuals and passive particles due to diffusion is simulated as a random walk following Visser (1997). The following equation computes the change in position ($z_t$) for an individual particle from the depth at time $t$, to time $t + 1$ ($z_{t+1}$) over a finite time step, $\delta t$:

$$z_{t+1} = z_t + K'_v(z_t)\,\delta t + R\,\sqrt{2\,r^{-1}\,K_v(z_t + \tfrac{1}{2}\,K'_v(z_t)\,\delta t)\,\delta t}, \tag{35}$$

where $K'_v(z_t)$ represents the gradient of diffusivity ($= \delta K_v/\delta z$) at depth $z_t$, $R$ is a random factor corresponding to a uniformly distributed random variable with a mean of zero and variance $r$ ($r = \frac{1}{3}$ for a uniform distribution between -1 and 1) (Ross and Sharples, 2004).

We assume no vertical velocity of currents in the system. The sinking rate of phytoplankton cells depends on cell size, following the allometric relationship in Durante et al. (2019) (Table 2).

To ensure that the random walk works correctly, the Lagrangian module also computes the movements of 1000 passive inert particles whose trajectories can be tracked.





**Table 1.** Fixed parameters of the 1D-hybrid model.

| Parameter | Symbol | Value | Units |
|---|---|---|---|
| Phytoplankton maximal chlorophyll to nitrogen ratio[1] | $\theta_{\max}^N$ | 3 | g Chl mol N$^{-1}$ |
| Phytoplankton cost of biosynthesis[1] | $\zeta$ | 3.00 | mol C mol N$^{-1}$ |
| Shape-factor describing the dependence of $V_{\max}^C$ on $Q^{N}$[1] | $n$ | 1.00 | dimensionless |
| Phytoplankton maximal growth rate when $T_{\mathrm{opt}} = 15\,^{\circ}\mathrm{C}$[2] | $\mu'$ | 5 | d$^{-1}$ |
| Interspecific activation energy[2] | $E_i$ | 0.22 | eV |
| Normalization constant for $E_d$[2] | $E_{d0}$ | 2.3 | eV |
| Normalization constant for $E_a$[2] | $E_{a0}$ | 0.98 | eV |
| Activation energy for heterotrophic processes[3] | $E_z$ | 0.65 | eV |
| Boltzmann constant | $k_b$ | $8.62 \times 10^{-5}$ | eV K$^{-1}$ |
| Scaling exponent against $\theta$ for $E_a$[2] | $\beta$ | -0.20 | eV |
| Scaling exponent against $\theta$ for $E_d$[2] | $\phi$ | 0.27 | eV |
| Normalization constant of the $\sigma_{PSII}$ and $\theta^C$ relationship[4] | $\delta$ | 0.492 | m$^2$ W$^{-1}$ g C g Chl$^{-1}$ |
| Exponent of the $\sigma_{PSII}$ and $\theta^C$ relationship[4] | $\kappa$ | 0.469 | dimensionless |
| Turnover time of the electron transfer chain[4] | $\tau$ | $5.50 \times 10^{-3}$ | s |
| Damage constants of a PSU[4] | $k_d$ | $5.00 \times 10^{-6}$ | dimensionless |
| Zooplankton grazing half-saturation constant | $K_{P,j_{\mathrm{pred}}}$ | 0.15 | mmol N m$^{-3}$ |
| Coefficient of the prey refuge[5] | $\Lambda$ | -6.60 | (mmol N)$^{-1}$ m$^3$ |
| Zooplankton gross growth efficiency[6] | $\xi$ | 0.30 | dimensionless |
| Fraction of unassimilated food by zooplankton[6] | $\eta$ | 0.24 | dimensionless |
| Zooplankton mortality rate at 15 °C | $m_z$ | 0.005 | d$^{-1}$ |
| Standard deviation of zooplankton grazing kernel[5] | $\sigma_{j_{\mathrm{pred}}}$ | 0.5 | dimensionless |
| Conversion rate from detritus to DIN at 15 °C | $R_{dn}$ | 0.10 | d$^{-1}$ |
| Detritus sinking rate | $W_d$ | 0.50 | m d$^{-1}$ |
| Light attenuation coefficient for seawater[7] | $K_w$ | 0.04 | m$^{-1}$ |
| Light attenuation coefficient for chlorophyll[7] | $K_{\mathrm{Chl}}$ | 0.025 | (mg Chl m$^{-2}$)$^{-1}$ |

1. Geider et al. (1998); 2. Chen (2022); 3. Brown et al. (2004); 4. Nikolaou et al. (2016); 5. Ward et al. (2012); 6. Buitenhuis et al. (2010); 7. Gan et al. (2010).




**Table 2.** Size scaling coefficients of phytoplankton and zooplankton traits following the general formula ($y = aV^b$; $V$: cell volume ($\mu$m$^3$)). For the size scaling of maximal phytoplankton growth rate and respiration rate, *see* Eqs. 12 and 13.

| Variable | Symbol | a | b | Units |
|---|---|---|---|---|
| Phytoplankton cellular carbon[1] | $P_C$ | 0.20 | 0.88 | pmol C cell$^{-1}$ |
| Phytoplankton half-saturation constant for nitrogen uptake[2] | $K_N$ | 0.14 | 0.33 | mmol N m$^{-3}$ |
| Phytoplankton cellular maximum N:C ratio[3] | $Q_{max}^N$ | 0.25 | -0.07 | mol:mol |
| Phytoplankton cellular minimal N:C ratio[4] | $Q_{min}^N$ | 0.07 | -0.17 | mol:mol |
| Sinking rate of phytoplankton[5] | $W_{phy}$ | 0.0019 | 0.43 | d$^{-1}$ |
| Zooplankton maximum grazing rate[6] | $G_{max}$ | 21.90 | -0.16 | d$^{-1}$ |

1. Menden-Deuer and Lessard (2000); 2. Edwards et al. (2012); 3. Marañón et al. (2013); 4. Ward et al. (2012); 5. Durante et al. (2019)

### 2.7 Initial conditions

Initial concentrations of DIN were interpolated from the January profile of BATS from the World Ocean Atlas (Garcia et al., 2024). The total initial concentration of all zooplankton size classes was assumed as 0.1 mol N $m^{-3}$ throughout the water column and was split equally among 20 size classes. We initialized 20000 phytoplankton super-individuals and 1000 passive particles and these numbers do not change during the simulation. The vertical positions of both the passive and the phytoplankton super-individual particles were randomly assigned between the surface (0 m) and the bottom (250 m) at the start of the simulation following a uniform distribution.

The ESD of each phytoplankton super-individual was randomly assigned between 0.80 and 60.00 $\mu m$ following a uniform distribution on the log space. The phytoplankton cellular carbon content was derived from cell volume following Marañón et al. (2013) (Table 2). The initial cellular nitrogen content was then estimated following the Redfield ratio (C:N = 106:16 mol:mol). The initial cellular chlorophyll content was estimated assuming a Chl:C ratio of 1:50 (g:g). The initial number of cells per phytoplankton super-individual was calculated assuming a uniform concentration of phytoplankton nitrogen of 0.1 mmol m$^{-3}$ throughout the water column.

### 2.8 Boundary conditions

To preserve total nitrogen, we apply Neumann boundary conditions for both surface and bottom boundaries. We also assume a reflective boundary for particles that encounter both surface and bottom boundaries during random walk. We leave the option to use the Dirichlet boundary condition if one wishes to.

### 2.9 Model simulations

To reach (quasi) steady-state seasonal cycles, the 1D-hybrid model was run for 6 years. Only the last year is analyzed in this study. We use the forward Euler method with a constant time step of 10 min throughout to numerically solve the differential



equations of biological processes. However, because the vertical random walk requires a short time step (Ross and Sharples, 2004), we allowed the particles to have 100 time steps within each biological time step (i.e., a time step of 6 s for the random
walk).

As the particle random walk is a significant step consuming computing time, we implemented parallel computing to run particle random walk using open MPI (Gabriel et al., 2004).

## 2.10  Observational data

For validating model outputs, we downloaded observational data of DIN (nitrate and nitrite), Chl, and NPP from the BATS
website (https://bats.bios.asu.edu/). To interpolate the data of each vertical grid at each date from the irregular cruise data, we used *k-nearest neighbours* (KNN) algorithm with each data point calculated as the mean of the three nearest neighbours.

## 3  Results

Below we first describe the behavior of the phytoplankton model, followed by an in-depth examination of the 1D model output. Then we compare the modelled fields of DIN, Chl, and NPP against observations to ensure that our model can at least
qualitatively reproduce the main patterns of observations. Afterwards, we show some patterns from the model outputs that may be interesting but hard to directly measure *in situ*.





## 3.1 Phytoplankton fitness landscapes

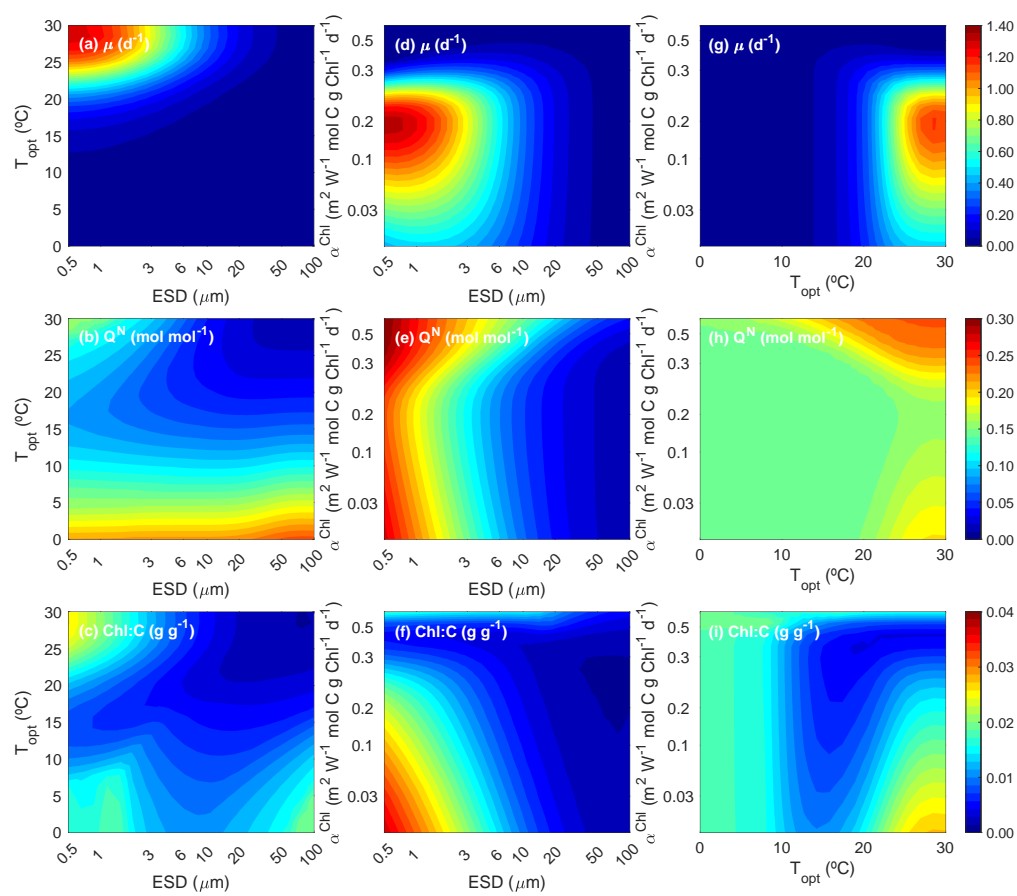

**Figure 3.** Contour plots for phytoplankton growth rate ($\mu$, d$^{-1}$), N:C ratio ($Q^N$, mol:mol) and Chl:C ratio (g:g) as functions of the three master traits (size, ESD ($\mu m$); optimal temperature, $T_{opt}$ (°C); and light affinity, $\alpha^{Chl}$ (W$^2$ m$^{-1}$ g Chl mol C$^{-1}$ d$^{-1}$)) at equilibrium under a typical summer condition (dissolved inorganic nitrogen = 0.10 mmol N m$^{-3}$, temperature = 28 °C, and PAR = 250 W m$^{-2}$).

Fig. 3 illustrates how growth rate ($\mu$, d$^{-1}$) (top row), $Q^N$ (mol:mol) (middle row), and the Chl:C ratio (g:g) (bottom row) vary for different phytoplankton ecotypes (i.e., a phytoplankton with a characteristic ESD calculated from $C_{div}$, $T_{opt}$ and $\alpha^{Chl}$).
To facilitate understanding the responses, we plot the 2D contour plots by varying two traits (i.e., ESD vs $T_{opt}$, ESD vs $\alpha^{Chl}$, $T_{opt}$ vs $\alpha^{Chl}$), while keeping the third trait constant. Below, we present the results for the summer period, characterised by a constant DIN concentration of 0.10 mmol m$^{-3}$, a temperature of 28 °C, and an irradiance of 250 W m$^{-2}$.

In the left column, we present the effects of size and $T_{opt}$, with a constant $\alpha^{Chl}$ of 0.08 W$^2$ m$^{-1}$ g Chl$^{-1}$ mol C d$^{-1}$. This $\alpha^{Chl}$ value is designed to adapt to the current light conditions without strong photoinhibition. As anticipated, the maximum





peak of $\mu$ is observed for small-size phytoplankton (< 3 $\mu$m), efficient at taking up nitrogen at low environmental concentrations, and with a $T_{opt}$ around 28-30 °C, close to the environmental temperature (Fig. 3a). Moreover, we observe close to zero $\mu$ values for the whole size range when $T_{opt}$ is far from the environmental temperature, highlighting maladapted ecotypes. $Q^N$ shows maximum values for small phytoplankton cells, almost regardless of $T_{opt}$ (Fig. 3b), decreasing towards largest sizes, consistent with the observations by Baer et al. (2017). In this particular scenario, where both irradiance and $\alpha^{Chl}$ are constant,

the Chl:C ratio also peaks for small cells with high $T_{opt}$ (Fig. 3c), similar to $\mu$. The increase in the Chl:C ratio for large-size ecotypes and $T_{opt}$ < 10 °C may be related to the minimum growth in carbon content (Fig. 3a).

In the middle column, we present how growth, $Q^N$ and Chl:C ratios change with size and $\alpha^{Chl}$, with a constant $T_{opt}$ of 30 °C. This $T_{opt}$ value is close to the environmental temperature (28 °C), indicating adapting to the ambient temperature. $\mu$ peaks for small phytoplankton (< 3 $\mu$m) and $\alpha^{Chl}$ around 0.10 W$^2$ m$^{-1}$ g Chl$^{-1}$ mol C d$^{-1}$ (Fig. 3d). The light condition of

250 W m$^{-2}$ favor organisms with low light affinity values ($\alpha^{Chl}$ < 0.20 W$^{-1}$ m$^2$ g Chl$^{-1}$ mol C d$^{-1}$) with high capability of photo-repair. $Q^N$ mainly depends on size, progressively decreasing towards the largest sizes (Fig. 3e). Other things being equal, the cells with more optimal $\alpha^{Chl}$ values tend to have lower N:C (Fig. 3e). Chl:C ratio shows the maximum values for the smallest phytoplankton with the lowest $\alpha^{Chl}$, and the minimum values for the largest phytoplankton with the highest $\alpha^{Chl}$ (Fig. 3f). As larger $\alpha^{Chl}$ increases the probability of photoinhibition and reduces the carbon-specific photosynthesis rate, the

amount of cellular chlorophyll content also decreases relative to carbon.

In the right column, we present the effects of $T_{opt}$ and $\alpha^{Chl}$, with a constant ESD of 1.36 $\mu$m. As anticipated, given the defined environmental conditions, the highest growth rate ($\mu$) is achieved by phytoplankton cells with a $T_{opt}$ close to the environmental temperature and $\alpha^{Chl}$ values around 0.10 W$^{-1}$ m$^2$ g Chl$^{-1}$ mol C d$^{-1}$, which are better adapted to high irradiance (Fig. 3g). Moreover, $Q^N$ increases with $T_{opt}$, with a minimum corresponding to cells with high growth (Fig. 3h).

Similarly, the Chl:C ratio is maximized for ecotypes with $T_{opt}$ closer to the environmental temperature, and characterized by lower $\alpha^{Chl}$ values, resulting in reduced photoinhibition (Fig. 3i).

## 3.2 Comparisons with observations

We compared the observed DIN, Chl, and NPP at the BATS station with the model output (Fig. 4). Our model is able to reproduce the general qualitative patterns of these three variables, albeit with some quantitative differences. Both the observation

and the model output show an increase in surface DIN and Chl during the winter when mixing is the strongest (Fig. 2). The model also successfully reproduces the deep chlorophyll maximum between 50 and 100 m. In spite of the presence of the deep chlorophyll maximum, the model can also reproduce the surface maximum of NPP which extends below 50 m during the summer. In other words, our model manages to resolve the paradox of low Chl but high NPP at surface waters during the summer in the oligotrophic subtropical ocean.

Admittedly, there are some differences between the model output and the observations. The model appears to underestimate DIN in deep waters and overestimate Chl in the euphotic zone, which is likely because the phosphorus limitation at BATS is not considered in our model (Ammerman et al., 2003).





**Figure 4.** Comparing (a, b) DIN (mmol m$^{-3}$), (c, d) Chl (mg m$^{-3}$), and (e, f) NPP (mg C m$^{-3}$ d$^{-1}$) between observations and model outputs.





### 3.3 Modelled patterns of passive particles

Fig. 5 compares the vertical distribution of passive particles and phytoplankton super-individuals, enabling us to verify if the
Lagrangian module properly models the random walk of particles. While the distribution of phytoplankton super-individuals
can be additionally affected by cell division and death, the distribution of passive particles is driven purely by diffusion.

As passive particles (Fig. 5a) are homogeneously distributed throughout the water column, we can affirm that the particle
random walks are working correctly. During periods of high vertical mixing, the distribution of super-individuals shows a
homogeneous pattern (Fig. 5b). However, as expected, during the period of strong stratification of the water column (April
- October), more pronounced differences were observed. A greater number of super-individuals were found in the euphotic
layer, where conditions for survival are optimal, with their abundance decreasing in deeper layers.

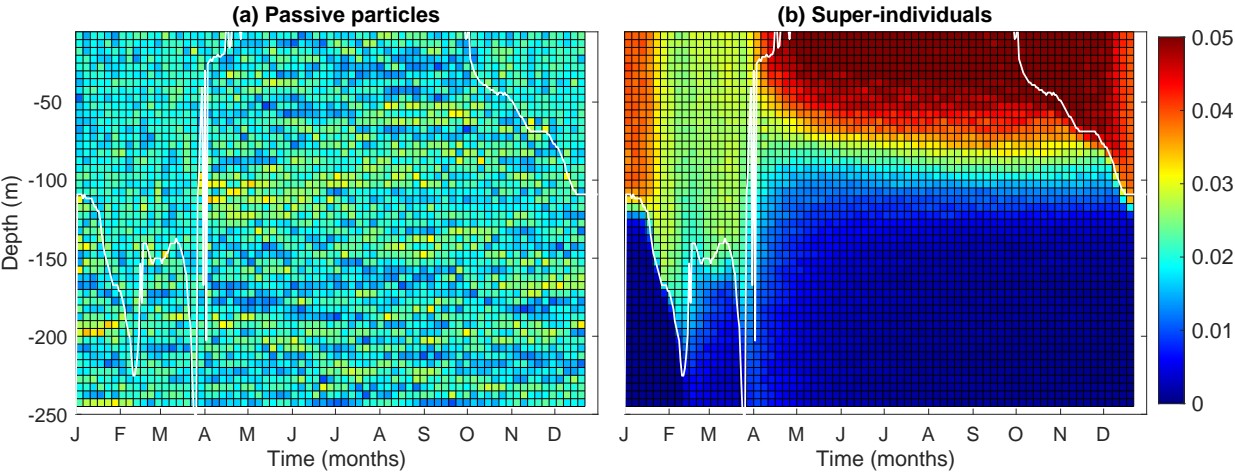

**Figure 5.** Mean temporal and vertical frequencies of (a) passive particles and (b) phytoplankton super-individuals every 5 m depth and every
5 days along the last year's simulation. The white lines represent the Mixed Layer Depth (MLD, m).

### 3.4 Modelled patterns of phytoplankton carbon, nitrogen, zooplankton and detritus

Phytoplankton carbon and nitrogen concentrations, for which no observational data are available, show similar patterns. Both
of them peak at spring after the shoaling of mixed layer depth (Fig. 6A,B). Before the spring bloom, surface phytoplankton
carbon and nitrogen can penetrate deeper than 150 m due to strong winter mixing. Similar to the patterns of chlorophyll,
phytoplankton carbon also shows elevated concentrations at the deep chlorophyll maximum during summer, although this
maximal layer is not as pronounced as that of chlorophyll.

By contrast, zooplankton biomass peaks in the summer and autumn after the phytoplankton spring bloom (Fig. 6C), sug-
gesting that the demise of the phytoplankton spring bloom is at least attributed to intensified zooplankton grazing. Detritus also



peaks after the phytoplankton spring bloom, but unlike zooplankton, its concentration gradually declines with time, as a result
of both accelerated decomposition due to high temperature and shifts in zooplankton size structure.

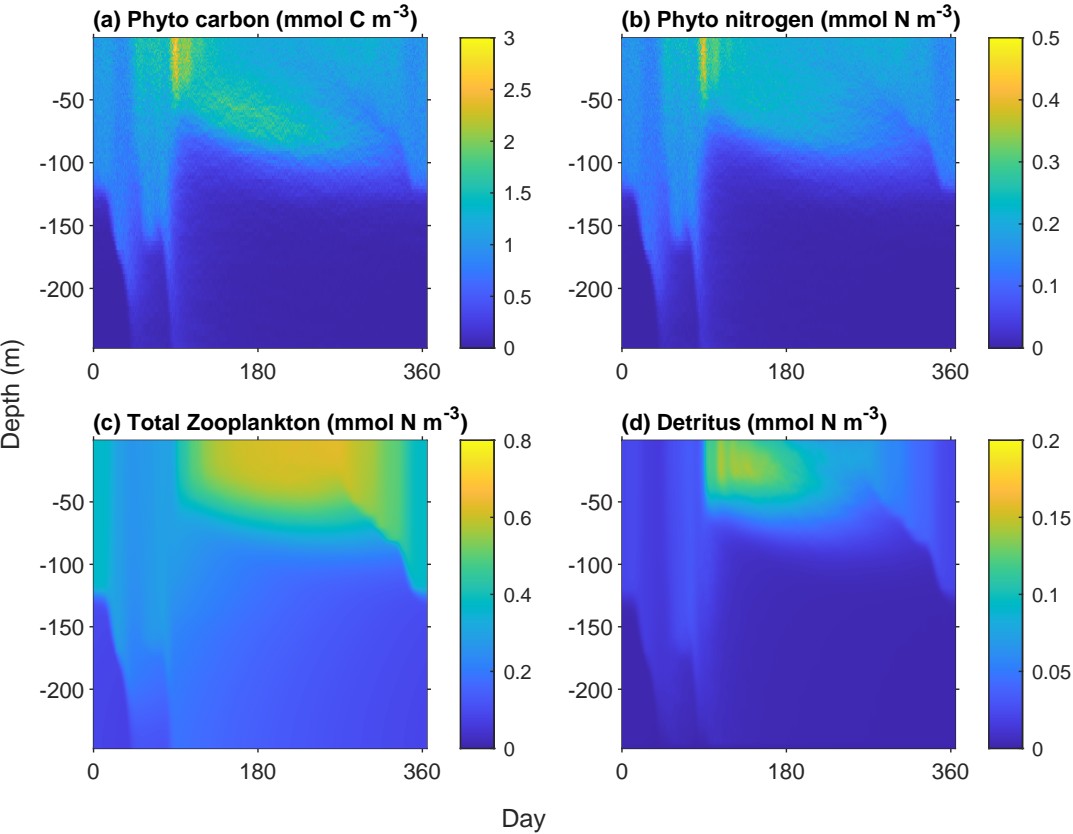

**Figure 6.** Modelled patterns of (a) phytoplankton carbon, (b) phytoplankton nitrogen, (c) total zooplankton nitrogen, and (d) detritus.

### 3.5    Modelled Chl:C and N:C ratios

The phytoplankton cellular N:C and the Chl:C ratios are important phytoplankton properties that show the extent of how
phytoplankton cells acclimate to the light and nutrient environment. They are also crucial to link phytoplankton carbon to
satellite observations of chlorophyll to and to the limiting element nitrogen (Fig. 7).

The model output provides higher Chl:C ratios in the surface mixed layer during the winter as a result of enhanced nutrient
supply due to strong mixing. The Chl:C ratios are low in both the surface layer in the summer and in deep waters beneath the
surface mixed layer. These patterns can be understood as an outcome of the balance between photosynthesis and chlorophyll
synthesis. Under high light and low nutrient conditions, phytoplankton cells reduce the rate of chlorophyll synthesis relative to
carbon synthesis and vice versa. However, when the light is too low, the synthesis rate of chlorophyll is too low to sustain the
maintenance of chlorophyll, leading to a phenomenon known as "bleaching" (Pahlow et al., 2013; Behrenfeld et al., 2016).



While the patterns of Chl:C ratios can be easily understood from the perspective of environmental control, N:C ratios are more related to changes in phytoplankton size than to the environment DIN or light. While one might expect that phytoplankton cells should have lower N:C at the surface during the summer due to low DIN, the model actually predicts the opposite pattern.

This is because the summer surface waters are dominated by small cells which tend to have larger N:C ratios (Marañón et al., 2013; Baer et al., 2017).

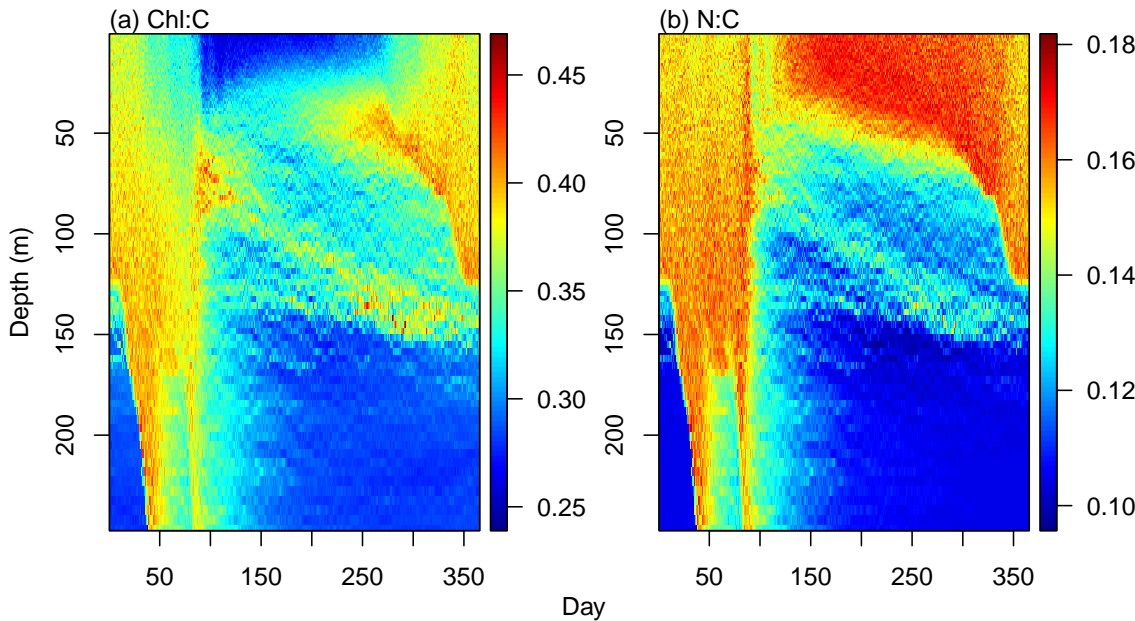

**Figure 7.** Modelled patterns of phytoplankton (a) Chl:C ratio (g Chl mol $C^{-1}$) and (b) N:C ratio (mol N mol $C^{-1}$).

### 3.6    Modelled phytoplankton trait distribution and functional diversity

Our model shows a typical pattern that phytoplankton mean size increases with nutrient availability (Fig. 8A). Phytoplankton mean size is the smallest at the surface during the summer and autumn where DIN is low, but increases with depth as nutrients

become more abundant. During winter, phytoplankton mean size is also larger at the surface during the winter (but not as large as in deeper waters during summer) when nutrient concentrations are higher due to stronger mixing.

   Phytoplankton size variance ($\mathrm{Var}(C_{div})$) is an index for size diversity and is the greatest in the area of deep chlorophyll maximum during summer and autumn and the lowest beneath the euphotic zone (150 m) (Fig. 8B). The high size variance at deep chlorophyll maximum results from the movements of small cells from above and large cells from below. Size variance is

also slightly greater at the surface during the winter than the summer due to more abundant nutrients, but not as high as those in the deep chlorophyll maximum. This suggests that dispersal probably plays the most important role in affecting size diversity.





As expected, phytoplankton community mean $T_{opt}$ largely follows seawater temperature, with higher values at the surface during summer and lower at depth (Fig. 8C). However, phytoplankton mean $T_{opt}$ show the lowest values at the area of deep chlorophyll maximum during summer and autumn, corresponding to the maximal variance of $T_{opt}$ (Fig. 8D). Otherwise, the

variance of $T_{opt}$ generally shows larger values in the surface mixed layer than at depth.

Phytoplankton mean light affinity ($\alpha^{Chl}$) also follows the opposite pattern of light availability, being the lowest at the summer surface and the highest in the deepest waters (Fig. 8E). The variance of $\alpha^{Chl}$ shows a qualitatively similar pattern with those of variance of $C_{div}$ and $T_{opt}$, being higher during the winter and at the deep chlorophyll maximum (Fig. 8F), again suggesting mixing can enhance trait variance and diversity.

The covariances between traits suggest selection forces against different traits. The covariance between $T_{opt}$ and $C_{div}$ is largely negative during the winter, suggesting larger cells tend to be cold-adapted at local scales. Conversely, Cov($T_{opt}$, $C_{div}$) is often positive at surface and deep chlorophyll maximum during summer at local scales. It is important to note that these covariances are calculated at local scales. We can still observe negative Cov($T_{opt}$, $C_{div}$) for the whole water column in which the small and warm-adapted cells are at the surface and large and cold-adapted cells are at depth.

The covariance between $T_{opt}$ and $\alpha^{Chl}$ is largely negative, suggesting that due to the negative environmental correlation between temperature and light, cold-adapted cells tend to adapt to the low light. The covariance between size and $\alpha^{Chl}$ is mostly positive during winter and at deep chlorophyll maximum, indicating larger cells tend to adapt to low light.





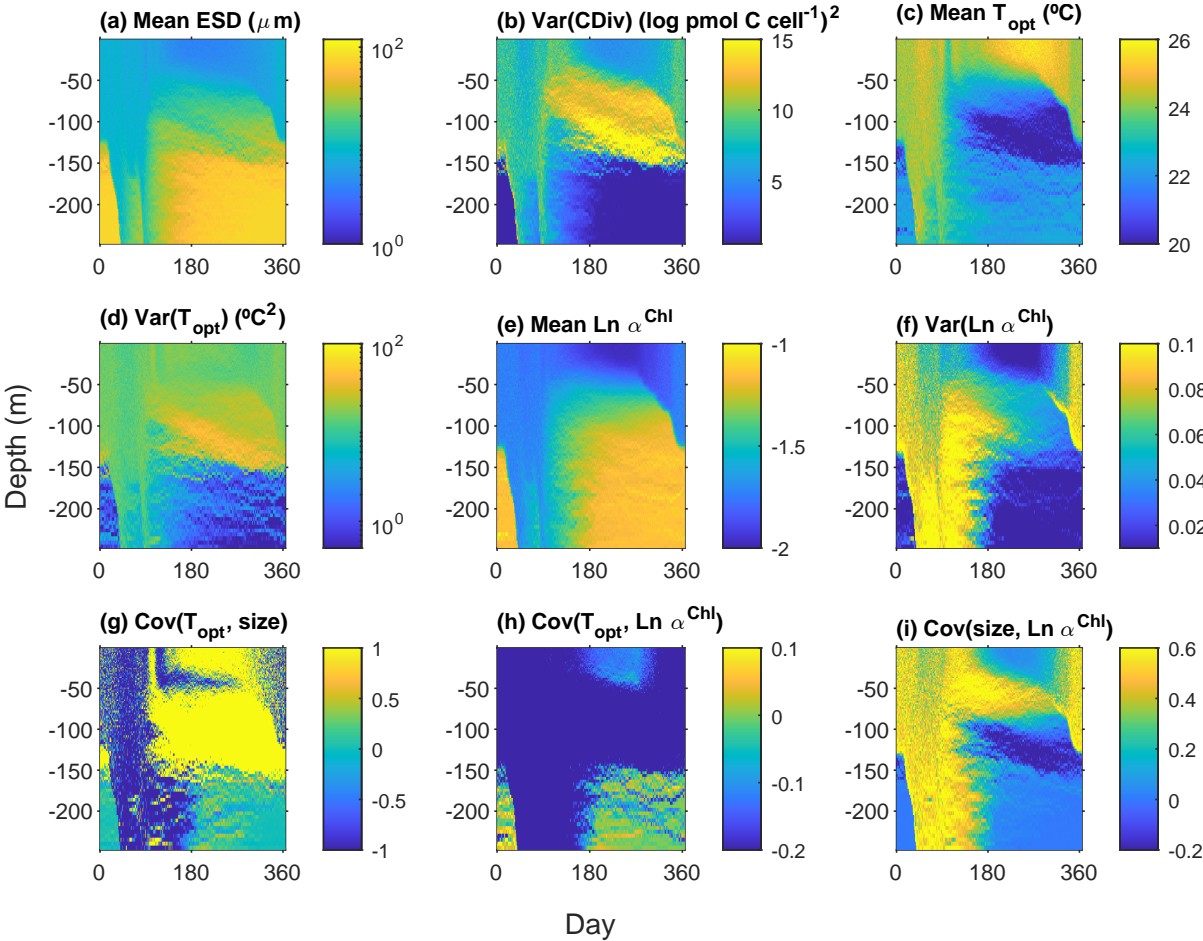

**Figure 8.** Temporal and vertical patterns of traits weighted by phytoplankton carbon content (Eq. 14). (A) Mean phytoplankton size (ESd) back-transformed from the carbon threshold of cell division ($C_{div}$). (B) Variance of phytoplankton $C_{div}$. (C) Mean phytoplankton optimal temperature ($T_{opt}$). (D) Variance of phytoplankton $T_{opt}$. (E) Mean phytoplankton light affinity represented by ln-transformed slope of Photosynthesis-Irradiance curve ($\alpha^{Chl}$). (F) Variance of phytoplankton Ln $\alpha^{Chl}$. (G) Covariance between phytoplankton $T_{opt}$ and Ln $C_{div}$. (H) Covariance between phytoplankton $T_{opt}$ and Ln $\alpha^{Chl}$. (G) Covariance between phytoplankton Ln $C_{div}$ and Ln $\alpha^{Chl}$.

The modelled patterns of functional richness largely follow those of (co)variances (Fig. 9). Both functional richness and evenness are greater during the winter season with strong mixing and at the layer of deep chlorophyll maximum.



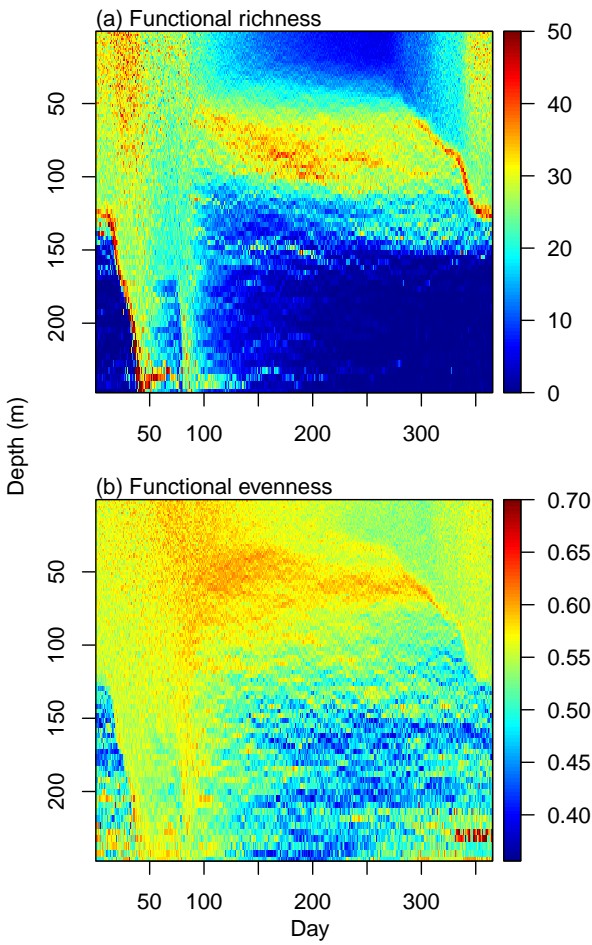

**Figure 9.** Modelled patterns of phytoplankton functional (a) richness and (b) evenness weighted by cellular carbon.

## 3.7 Plankton size spectra

During both seasons, log abundances form linear relationships with log size (biovolume) for both phytoplankton and zoo-plankton. The slopes of phytoplankton size spectra were between -1.35 and -1.1, consistent with what would be expected for phytoplankton communities in oligotrophic oceans (Fig. 10) (Marañón, 2019). The slopes of the phytoplankton size spectra were more negative in the summer than in the winter, also consistent with previous observations that the phytoplankton size spectra became steeper and small phytoplankton became more dominant when nutrient supply diminishes (Huete-Ortega et al., 2014).



By contrast, the slopes of zooplankton were similar between summer and winter and were less steep than those of phyto-
plankton (Fig. 10). This relates to three factors. First, large zooplankton have a wider feeding kernel than small ones, thus
having access to a wide range of prey. Second, the predator-prey size ratio increases with predator size, which makes the slope

of size spectra less steep (Trebilco et al., 2013). Third, large zooplankton can also feed on small zooplankton. These patterns
are consistent with the observations in marine plankton communities that more biomass can exist in larger size classes, which
leads to an inverted biomass pyramid (Gasol et al., 1997; Trebilco et al., 2013).



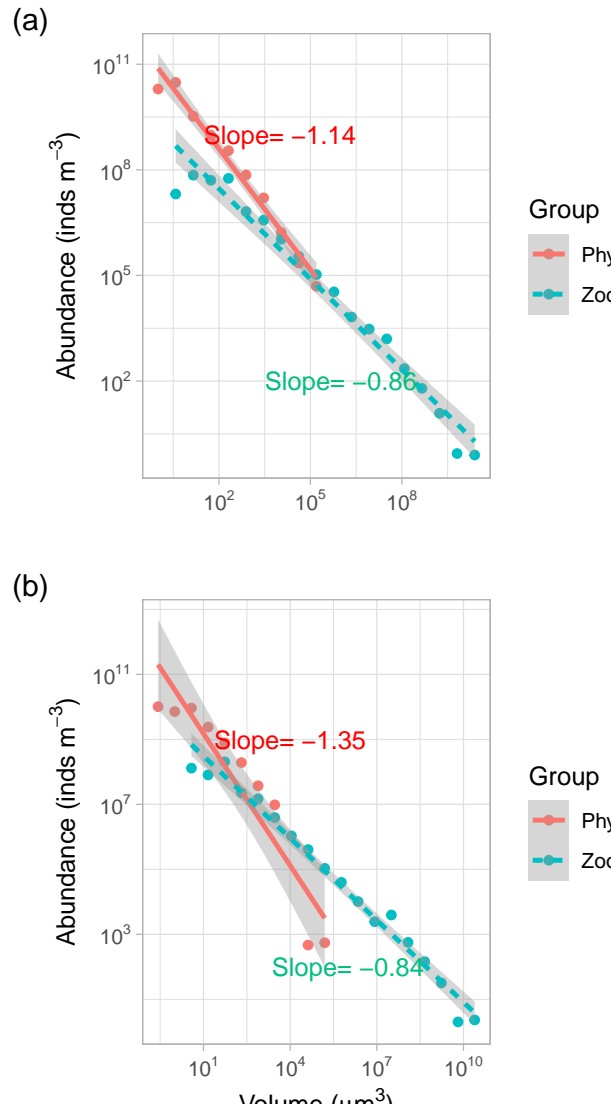

**Figure 10.** Phytoplankton (Phy) and zooplankton (Zoo) size spectra at the surface during the (a) Winter ($31^{st}$ March) and (b) Summer period ($31^{st}$ August). The dots show the abundance of each size class. The straight lines show the OLS regression lines. The slopes of each regression line are also shown.

## 3.8  Diel variability of phytoplankton cell

Our model allows to track the properties of a phytoplankton cell throughout a diel cycle to gain insights from the life history of the cell. Fig. 11 shows the trajectory of a randomly selected super-individual during the first week of the winter and the first week of the summer period (hourly resolution).





During the winter period, this phytoplankton particle was dispersed widely between the surface and 50 m in the water column due to strong mixing (Fig. 11a). Because the water column was well mixed, it was exposed to relatively stable conditions of temperature ($\sim$ 19.50 °C) and DIN (Fig. 11b,d). By contrast, this particle experienced variable PAR conditions, oscillating between daily maximums $\sim$ 50 W m$^{-2}$ and 200 W m$^{-2}$. By comparison, the variations of Chl:C ratio ($\theta^C$) are modest (Fig. 11h), suggesting its limited acclimation capability.

During the summer period, the super-individual position was more stable at around 35 m in the first seven days due to weak mixing (Fig. 11a). As a consequence, the environmental conditions (temperature, DIN, and PAR) the particle experienced were relatively stable. However, the variabilities of phytoplankton $Q^N$ and $\theta^C$ are of the same magnitude between summer and winter, mainly due to diel changes of light.

The phytoplankton $Q^N$ and $\theta^C$ were higher during the winter period (Fig. 11g, red line) than during the summer (blue line), reflecting the higher nutrient and lower light environment in the winter.

In spite of the seasonal differences, both particles divided twice during this period (Fig. 11e-f). The phytoplankton cellular carbon content increased progressively with time before the division event until reaching the division threshold (Fig. 11f). When this threshold was reached, the number of phytoplankton cells doubled and the cellular carbon, nitrogen and Chl content halved (Fig. 11e-f).

Irrespective of the birth event, we can also observe clear diel changes in cellular carbon, nitrogen, and chlorophyll contents induced by light-driven photosynthesis, nutrient uptake, chlorophyll synthesis, and dark respiration. Cellular carbon increased from sunrise to sunset but declined in the dark due to respiration. Correspondingly, N:C and Chl:C ratios declined during the daytime and increased during nighttime.

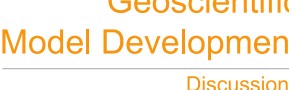
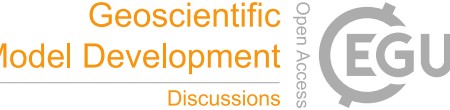

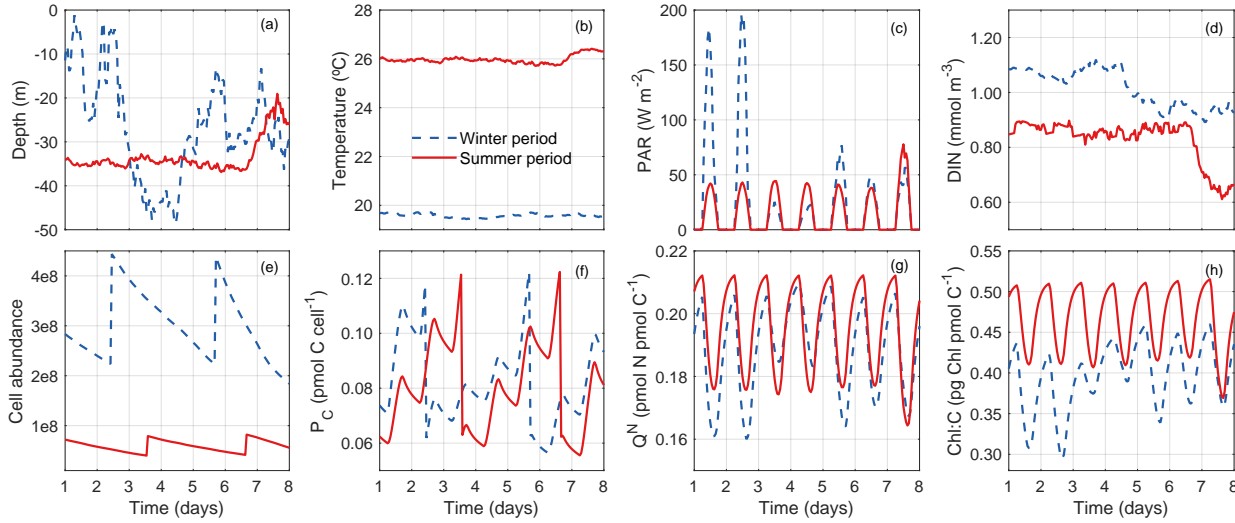

**Figure 11.** Tracking a randomly selected phytoplankton super-individual for 7 days (hourly resolution) during the winter (red solid lines) and the summer (blue dashed lines) periods. (a) Depths (m) between which the super-individual oscillated in the water column. (b) Differences in the Temperature ($^\circ$C) conditions experienced by the super-individual. (c) Differences in the Photosynthetically Active Radiation conditions (PAR; W m$^{-2}$) experienced by the super-individual. (d) Differences in the nitrogen concentration conditions (NO$_3$, pmol N m$^{-3}$) experienced by the super-individual. (e) Temporal evolution of phytoplankton cell abundance (number of cells) within the selected super-individual. (f) Temporal evolution of cellular carbon content ($P_C$, pmol C cell$^{-1}$). (g) Temporal evolution of nitrogen cellular quota ($Q^N$, pmol N pmol C$^{-1}$). (h) Temporal evolution of chlorophyl to carbon cellular ratio ($\theta^C$, mg Chl mmol C$^{-1}$). Vertical dashed lines indicate the beginning of each day. (The panel labels are missing).

## 3.9 Effect of number of super-individuals

We investigated the effect of the number of phytoplankton super-individuals on the key model outputs such as DIN, phytoplankton biomass and diversity by running three simulations with 5,000, 10,000, and 20,000 super-individuals. We computed the temporal trends of key variables integrated throughout the water column during the final year of each simulation (Fig. 12).

The number of phytoplankton super-individuals has negligible effects on the bulk properties such as DIN, ZOO, $P_N$, $P_C$, Chl and NPP (Fig. 12a-f). However, the number of phytoplankton super-individuals has noticeable effects on phytoplankton functional richness and evenness. The lower numbers of super-individuals lead to lower estimates of phytoplankton functional richness and evenness than 20,000 super-individuals during most times of the year. Moreover, the simulation with lower numbers of super-individuals struggles to converge to regular seasonal cycles of functional richness and evenness, suggesting

that a sufficient number of phytoplankton super-individuals is required to represent phytoplankton diversity.



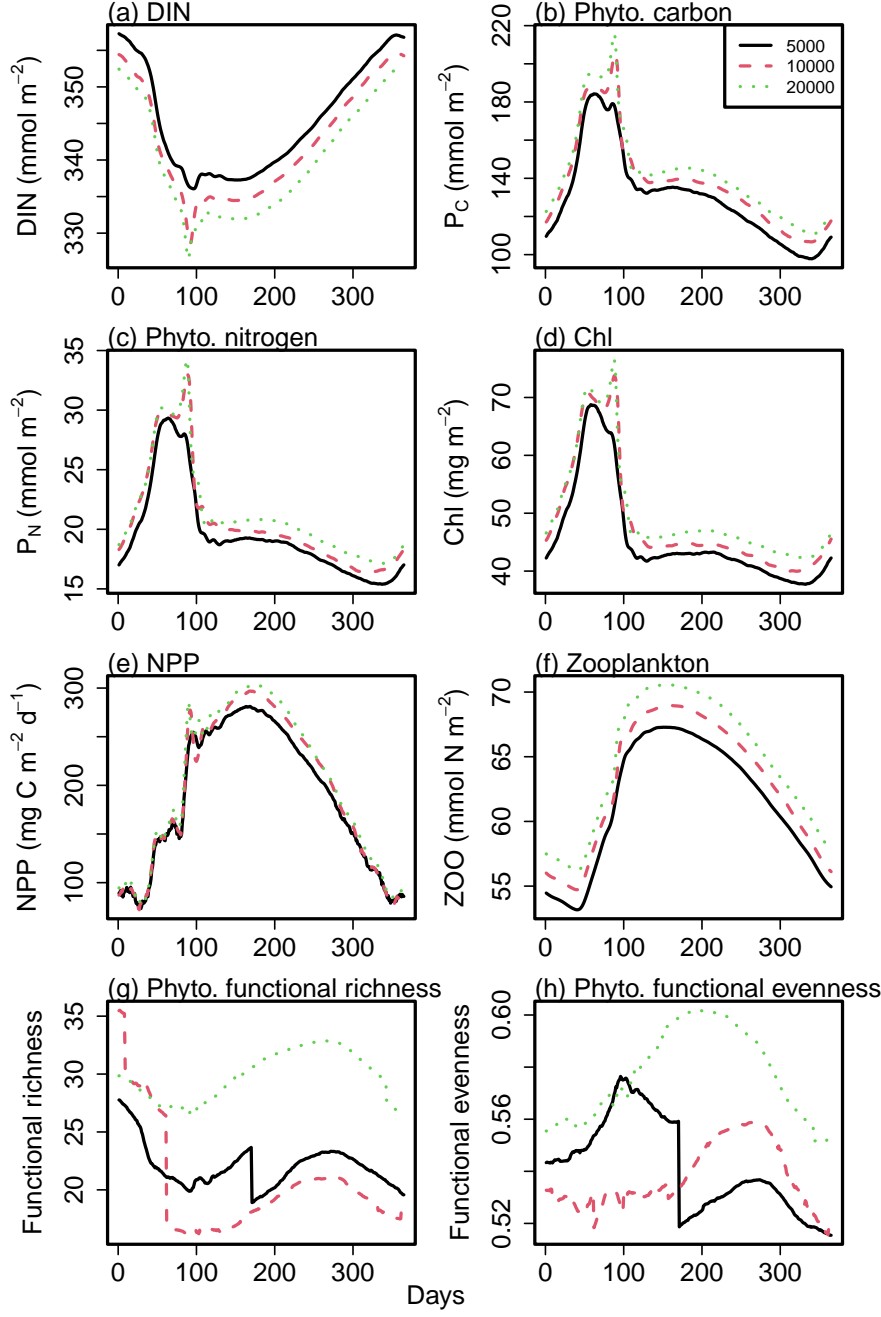

**Figure 12.** Comparisons of simulations with different numbers of super-individuals on vertically integrated variables. (a) Dissolved inorganic nitrogen (DIN, mmol m$^{-2}$). (b) Phytoplankton carbon ($P_C$, mmol C m$^{-2}$). (c) Phytoplankton nitrogen ($P_N$, mmol N m$^{-2}$). (d) Chlorophyll (Chl, mg m$^{-2}$). (e) Net primary production (NPP, (mg C m$^{-2}$ d$^{-1}$). (f) Zooplankton biomass (ZOO, mmol N m$^{-2}$). (g) Phytoplankton functional richness. (h) Phytoplankton functional evenness.





## 4 Discussion

We have presented a novel hybrid Eulerian-Lagrangian plankton model that treats phytoplankton as particles or superindividuals. We assign three master traits (cell size ($C_{div}$), temperature affinity ($T_{opt}$), and light affinity ($\alpha^{Chl}$)) to each phytoplankton super-individual and these master traits determine many other physiological traits involved in nutrient uptake and photosyn-

thesis. We also incorporate acclimation and evolution in this model, making it suitable for addressing many questions in phytoplankton ecology and evolution.

### 4.1 A brief history of individual-based models on phytoplankton

In recent decades, individual-based models have gained popularity in ecology modelling. This approach allows to simulate the non-linear effects of the physical environment (Woods and Onken, 1982), the interaction between individuals, the life

cycles of individuals (Grimm et al., 2006; Hellweger et al., 2008; Hense and Beckmann, 2010), the adaptive behaviour, and the intrapopulation variability in response to internal and external environmental conditions (Hellweger et al., 2008; Grimm et al., 2010).

In oceanography, the first individual-based models were developed between the late 1970s and early 1980s (Ledbetter, 1979; Platt and Gallegos, 1981; Falkowski and Wirick, 1981; Woods and Onken, 1982). However, because of their high computational

cost, it was not until recent decades that they started to gain momentum (Cianelli et al., 2004; Woods, 2005; Nogueira et al., 2006; Cianelli et al., 2009; Beckmann et al., 2019; Ranjbar et al., 2021). To obtain a realistic representation of the phytoplankton community, it is necessary to model a large number of Lagrangian particles. Each particle is modelled individually, thus obtaining a unique history of interaction with the environment and other particles. Therefore, the computational cost will be proportional to the number of particles considered. One way to limit this computational cost is to model Lagrangian particles

as super-individuals (Scheffer et al., 1995), although this approach may restrict the population's heterogeneity and diversity (Ranjbar et al., 2021) if the total number of particles is not high enough. With the increasing computing power, this problem is of less severity in individual-based models.

Several studies, employing the Lagrangian Ensemble model, have also depicted zooplankton as Lagrangian particles (agents) (Woods and Barkmann, 1993, 1994; Woods, 2005; Nogueira et al., 2006). However, these studies define a singular class of

herbivorous zooplankton that feeds indiscriminately on all phytoplankton particles, irrespective of their size. Furthermore, they do not account for the interaction between a phytoplankton particle and zooplankton, essentially adopting an Eulerian approach. In other words, when calculating grazing, a zooplankton particle is allowed to consume phytoplankton particles within the same grid layer without considering physical interaction. While our model has considered zooplankton size structure and size-dependent feeding kernels, it will be our next step to model zooplankton as Lagrangian particles which can account

for more detailed interactions between zooplankton and phytoplankton particles.

In summary, individual-based models are gaining popularity thanks to the increasing computing power and they can provide novel insights that conventional models fail to provide.



## 4.2 Strengths and limitations of the model

There are several strengths of our model. First, the individual-based phytoplankton model captures the realistic physical move-
ment and acclimation status of a phytoplankton cell affected by ocean turbulence. It has been widely acknowledged that for
both Eulerian models and *in situ* incubations, the estimates of primary production can be biased although the extent of this
bias is uncertain (Barkmann and Woods, 1996; Ross and Geider, 2009; Baudry et al., 2018). Part of the differences in the bias
estimates may be due to the traits of the phytoplankton cells. Most of the previous individual-based phytoplankton models
do not consider phytoplankton functional diversity (i.e., the different traits associated with each cell). While the aim of this
paper is not to compare the primary production estimates between the Eulerian model and Lagrangian model (as developing an
Eulerian version of the same plankton model is beyond the scope of this paper), it remains to be investigated how the estimates
of primary production can differ between the Eulerian model and Lagrangian model in different environments (e.g., stratified
vs. well-mixed water column).

Second, our phytoplankton model captures three dimensions of phytoplankton traits. We have followed the DARWIN model
(Follows et al., 2007; Barton et al., 2010; Dutkiewicz et al., 2020) to assign three master traits to phytoplankton albeit with
slight differences. The assumption of designing these three traits is that they are largely orthogonal to each other. In other
words, small and large cells have an equal probability of being warm-adapted or cold-adapted and they also have the same
probability of being high-light-adapted or low-light-adapted. How the changes of these traits affect phytoplankton acclimation
and the differences in primary production estimates between Eulerian and Lagrangian models remain to be investigated using
our model.

Third, our phytoplankton model allows phytoplankton evolution. We have built the functionality of phytoplankton evolution
in the Lagrangian model enabling the mutation of all three master traits of phytoplankton super-individuals. Modelling phyto-
plankton evolution has been a hot topic recently (Ward et al., 2019; Beckmann et al., 2019) and the individual-based model is
an ideal and straightforward approach to accommodate mutation and evolution (Acevedo-Trejos et al., 2022). We will use this
model to further explore how evolution affects phytoplankton diversity and primary productivity.

We also highlight several limitations of our model which will be addressed in future work. One weakness of the current
model is the slow computation. To obtain a realistic representation of the phytoplankton community, it is necessary to model a
large number of particles. As each particle is modelled individually, the computational cost will be proportional to the number
of particles considered. To obtain a satisfactory model result of phytoplankton diversity, it is desirable to model as many
particles as possible (Fig. 12), thus incurring heavy computation costs.

The short time step is needed for correctly simulating the random walk (Ross and Sharples, 2004). In fact, to meet the
requirement of short timestep (<6 s) in the random walk but to minimize computation time, we have made the timestep of
biological reactions 100 times longer than the time step of the random walk.

We have also implemented openMPI parallel computing for simulating the random walk of both the passive and phytoplank-
ton particles. However, the computation is still too slow to allow effective sensitivity analysis or parameter optimization. In
the future, we will make the computations of biological reactions in parallel and will attempt openMP which may allow more





efficient memory sharing among threads. Another more advanced remedy would be to implement GPU computing to speed up the computation.

The second limitation relates to the inadequate parameterization of the phytoplankton model using laboratory data and the need for more extensive validation of the overall 1D model output. While the size scaling of nutrient uptake has been studied extensively in the literature (Edwards et al., 2012; Marañón et al., 2013) and the temperature dependence of phytoplankton growth seems clear (Thomas et al., 2012; Chen, 2022), how the light traits of phytoplankton affect growth are relatively unknown particularly when considering the effect of photoacclimation. Edwards et al. (2015) analysed the relationships between light traits (e.g., slope of the P-I curve). However, the model they used did not consider the dependence of photosynthetic rate on the Chl:C ratio or photoinhibition. To consider both the effects of Chl:C ratio and photoinhibition on photosynthesis of phytoplankton, we have to assume an empirical relationship between $\alpha^{Chl}$ and repair rate based on the contrast between high-light adapted and low-light adapted ecotypes of *Prochlorococcus* (Moore et al., 1998). However, it remains unclear how this scaling relationship can be applied to phytoplankton in general. We need more data on this to obtain a more reliable relationship between the photosynthetic parameters.

The model outputs of phytoplankton traits also need to be validated against observations. While the measurements of phytoplankton size structure can be obtained, other traits such as $T_{opt}$ and $\alpha^{Chl}$ are difficult to measure *in situ* on a cellular basis. Moreover, even the bulk properties such as the N:C and Chl:C ratios of the whole phytoplankton assemblage are difficult to measure *in situ* due to the difficulties associated with measuring phytoplankton carbon and nitrogen. Only a few studies have managed to measure cellular carbon and/or nitrogen of phytoplankton using flow cytometric sorting (Graff et al., 2012; Baer et al., 2017), while, for larger cells, most studies relied on microscopic counting to estimate phytoplankton cell volume which can then be converted to carbon (Cloern, 2018) without any measurement of cellular nitrogen. These types of information is essential for studying biogeochemical cycling and validating ecosystem model outputs.

Another limitation is that the model may be overly complicated if we want to understand the key factor in controlling some ecological phenomenon (*see* below). The model has included multiple traits and processes which form an intertwined feedback network that makes it challenging to isolate the direct effect of a single factor. For example, if we aim to assess whether primary production is limited by nutrient supply or light availability by performing a single-factor perturbation experiment, the increase in nutrient supply will not only affect the nutrient status but also the trait distribution of the whole phytoplankton community. This change in the mean trait of phytoplankton depends on the existing trait diversity of the community (and the mutation rate of individual cells) about which we have little information (Acevedo-Trejos et al., 2015; Chen et al., 2019). If a user already knows that the system can be simplified (e.g., there is little variability in phytoplankton thermal or light traits), the user can modify the initial condition and the mutation rate to remove the unnecessary trait variance. Thus, one can simplify this model to a single-trait (e.g., size) phytoplankton model if desired.

Finally, needless to say that our model only considers one limiting nutrient – nitrogen without considering other important elements such as phosphorus or silicate, which probably leads to some discrepancy between observations and our model outputs at BATS where phosphorus is limiting. It is up to the user to decide whether to add these nutrients to the model and also depends on the question being asked.





### 4.3 Potential applications

Below we discuss several potential applications of our model. Note that these are not exhaustive, but just serve as potential interesting directions.

#### 4.3.1 Validating *in situ* measurements of primary production

An obvious application of our model is to check the bias in *in situ* measurements of primary production for which incubation bottles are tethered at fixed depths and the phytoplankton cells within the bottles experience different light environments from those being mixed throughout the water column. While several studies have attempted to address this problem using Lagrangian phytoplankton models (Barkmann and Woods, 1996; Baudry et al., 2018; Tomkins et al., 2020), they often overlook

phytoplankton traits. The most important phytoplankton trait for this problem would be likely light-related traits (e.g., $\alpha^{Chl}$) which are rarely considered in phytoplankton Lagrangian models.

In addition, it is not only the trait itself but also the trait distribution that can matter for primary production. In other words, phytoplankton diversity and community composition have to be taken into account to assess how accurate the *in situ* measurements of primary production are. Since phytoplankton trait distribution is not static across time and space, the bias

also depends on the phytoplankton community being sampled, so there is no guarantee that the bias can be easily extrapolated to other cases. The caveat is that we need to know the phytoplankton trait distributions (which are even harder to measure) to assess the accuracy of primary production estimates.

#### 4.3.2 Understanding what controls phytoplankton diversity

The central theme of ecology revolves around understanding the factors that regulate biodiversity. Vellend and Agrawal (2010)

presented a unified view of four processes controlling biodiversity: selection, dispersal, drift, and evolution. While many studies investigate what regulates phytoplankton diversity in the ocean (Barton et al., 2010; Vallina et al., 2014; Righetti et al., 2019; Dutkiewicz et al., 2020), few managed to examine the holistic effects of all four processes on biodiversity.

Our model already incorporates the processes of selection, dispersal, and evolution and can be easily adapted to integrate drift to address this gap. One challenge would be again about the computational costs if one wishes to understand the large-

665 scale patterns of phytoplankton diversity by coupling the individual-based model with a global circulation model (Hellweger et al., 2014). Nevertheless, this challenge can be addressed using advanced computing techniques as described above.

#### 4.3.3 Understanding how phytoplankton acclimation and trait distribution affect phytoplankton distribution

Biological oceanographers have long been fascinated by distinctive patterns of phytoplankton distribution in the ocean such as the deep chlorophyll maximum (Cullen, 2015) and spring bloom (Behrenfeld, 2010; Lévy, 2015). Despite extensive study,

debates persist on what mechanisms drive these patterns.

The formation of deep chlorophyll maximum remains enigmatic with ongoing debate as to whether it reflects an actual accumulation of phytoplankton biomass or is primarily driven by photoacclimation for which phytoplankton cells increase





their intracellular pigment content to acclimate to the low light condition. There is evidence that some phytoplankton species may be more abundant at the surface but show a peak of total pigments at the layer of deep chlorophyll maximum, while some other species may indeed show greater biomass at this layer (Chen et al., 2011). Again, the location of the phytoplankton biomass peak likely depends on phytoplankton traits. Our model is an ideal tool for elucidating the contribution of real biomass versus the photoacclimation effect to the layer of deep chlorophyll maximum.

Similar arguments can be raised for the spring bloom. Behrenfeld (2010) argued that contrary to the light effects (i.e., the critical depth hypothesis), the coupling between phytoplankton growth and zooplankton grazing plays a critical role in inducing the spring bloom. Lévy (2015) also used a 1D NPZD model to test different hypotheses and highlighted the importance of physics forcing on the validity of each hypothesis. However, few studies have considered the roles of the changes in phytoplankton traits and photoacclimation in determining the onset of spring bloom. Our model presents an exciting opportunity to fill this knowledge gap.

### 4.3.4 Understanding diel variations in phytoplankton cell properties

Another potential application of our model is to understand the diel variations of cell size, abundance, and cell cycle of phytoplankton in the oligotrophic ocean (Vaulot et al., 1995; Li et al., 2022). It is an interesting phenomenon that picophytoplankton cells such as *Prochlorococcus* and *Synechococcus* tend to show synchronized growth over a diel light/dark cycle. Phytoplankton cells tend to increase the size from sunrise to sunset but divide in late afternoon or evening, thus creating a mismatch between cell carbon production and abundance (Li et al., 2022). However, these diel rhythms can be different for different groups of phytoplankton (Vaulot and Marie, 1999). It is still unclear how these different patterns can be entirely explained by environmental (light, nutrient) variations or at least partly due to endogenous circadian clock (Heath and Spencer, 1985; Hellweger et al., 2020).

As our model is driven by a diel light/dark cycle, it can be used to understand what regulates the changes in cell properties linked to the cell cycle. Our model can be further modified to include cell cycles of phytoplankton to understand what controls phytoplankton division (Pascual and Caswell, 1997). Another promising future direction is to include more molecular processes such as gene expression and protein synthesis into the phytoplankton cell cycle, thus allowing us to link molecular studies with phytoplankton traits (Hellweger, 2020; Hellweger et al., 2020).

### 5 Conclusions

We introduce a novel 1D-hybrid Eulerian-Lagrangian, uniquely tailored to explore how water column dynamics shape phytoplankton dynamics. Phytoplankton are modelled as super-individuals, a Lagrangian particle that represents a cluster of clonal phytoplankton cells that are physiologically identical and share a common history. Each phytoplankton super-individual is characterized by its cell size, temperature affinity and light affinity. Furthermore, these super-individuals possess the capability to mutate, enhancing the model's capacity to simulate phytoplankton growth, productivity, and diversity within dynamic aquatic environments.





The seasonal variability of temperature, irradiance, and vertical diffusivity at the BATS station enabled us to evaluate the response of our ecological model to environmental changes. By employing three master traits (size, temperature affinity, and light affinity), the individual-based model illustrates the evolution and adaptation of the phytoplankton community to environmental conditions and the competition between different phytoplankton ecotypes. Furthermore, the model allows individual analysis, allowing us to scrutinize how each phytoplankton super-individual responds to the environmental conditions it encounters

throughout its life cycle. The model also has several weaknesses, such as high computational costs, and the need for extensive parameterization, and validation. However, with the appropriate experimental design, it has several potential applications that would help us address questions related to the individual growth of phytoplankton, as well as the productivity and diversity of the phytoplankton community.

*Code and data availability.*  The model code and input data are publicly available at https://github.com/BingzhangChen/IBM

under the MIT license and also available on zenodo https://doi.org/10.5281/zenodo.13310586.

*Author contributions.*  BC conceived the study and wrote the model code. IS wrote the first draft of the paper. Both authors contributed to a later revision of the paper.

*Competing interests.*  The authors declare that they have no conflict of interest.

*Acknowledgements.*  We sincerely thank Sergio M. Vallina for the useful discussion and for sharing the eddy diffusivity data. This study

is supported by a Leverhulme Research Grant Project (grant number RPG-2020-389; P.I. BC). We also appreciate the support from the ARCHIE-WeSt High-Performance Computing Centre at the University of Strathclyde.



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
