# Peer review of "PIBM 1.0: An individual-based model for simulating phytoplankton acclimation, diversity, and evolution in the ocean"

_Geoscientific Model Development, 2024_

## Referee Comment (RC1)

**Review of 'PIBM 1.0: An individual-based model for simulating phytoplankton acclimation, diversity, and evolution in the ocean' by Sala and Chen**

The paper introduced a new individual-based plankton model that allows phytoplankton acclimation. The paper consists of a description of the model and a presentation of its results. This is a worthwhile effort that can contribute to the plankton modelling community. However, more effort seems to be needed to convey the meaning and structure of the model more accurately and to convince readers that this new model is an improvement over existing models in spite of substantially increased complexity. This is particularly true as the paper did not attempt to clarify phenomenon or compare it with the Eulerian model results.

Although the new model reproduces the seasonal variation of the underlying observed variables to some extent (Fig. 4), I do not expect that the results are an improvement over those from much simpler plankton models. The present model has an advantage of providing information that was not available before, but it is difficult to assess how reliable these results are (e.g., Fig. 3, 8, and 10). If validating a model directly is difficult, sensitivity tests can help. As many previous papers have predicted that the effect of phytoplankton acclimation is not important under the realistic ocean conditions, it is important to at least compare the results in which the effects of three phytoplankton traits (size, optimal temperature, and light affinities) are neglected. It is also interesting to see if the results without these effects differ from the Eulerian model results.

The model description is unclear, making it difficult to understand how the model is calcluated. I could not find the procedure for calculating the three traits in the paper. The proper explanation how these traits for Lagrangian plankton particles are calculated, except mutation (L194), and how their initial conditions are given, especially temperature and light affinity. I cannot understand the relationship between $C_{div}$, ESD, and V (in Table 2), while all these variables appear to represent the cell size. Nor can I understand how these traits are affected by the motion of Lagrangian particles such as $z_i$. Note that this is the most essential part of the new model.

It is important to mention that the inclusion of more empirical formulae does not necessarily improve the model. The authors tried to include a large number of existing empirical formulae without justifying why they should be included. The phytoplankton model largely follows that by Geider et al. (1998). Here it is important to specify how the new model differs from Geider et al. (1998). It is not clear whether the zooplankton model used the existing model, or developed a new model. Again, the model equations are listed without any explanation to justify their use.

Other points:

L.205: R package TPD - necessary to explain

L305: no explanation on how to calculate the size spectra of phytoplankton

L325: Information on the temporal resolution of forcing and on the type of the 1D model is necessary. If the 1D model includes nonlocal mixing, $K_v$ does not necessarily predict the vertical motion of particles.

L346: I should mention that the calculation of $z_i$ using random walk can exaggerate the variance of traits, because it neglects the correlation of two particles located nearby. It also neglects the effects of horizontal diffusion and patchiness.

L371: The Neumann boundary condition does not mean no flux at the boundary.

Fig. 2: It is unrealistic that MLD is zero during the summer.

Fig. 11: error in the caption (red solid lines for summer).

L547: This section should be moved to the introduction. It may also be important to include references of Lagrangian plankton models using different approaches such as Jokulsdottir & Archer (2016), Kida & Ito (2017), and Noh et al. (2021). It is especially so, when references of Lagrangian cloud models are included.

---

## Author Comment (AC1)

Dear Editor,

We sincerely thank you and the reviewers for the constructive comments provided. Please see below for our point-to-point responses.

Review 1

The paper introduced a new individual-based plankton model that allows phytoplankton acclimation. The paper consists of a description of the model and a presentation of its results. This is a worthwhile effort that can contribute to the plankton modelling community. However, more effort seems to be needed to convey the meaning and structure of the model more accurately and to convince readers that this new model is an improvement over existing models in spite of substantially increased complexity. This is particularly true as the paper did not attempt to clarify phenomenon or compare it with the Eulerian model results.

[Response] Thanks for the fair comment. We have made substantial efforts to revise the model and paper writing to better convey the meaning and structure of the model. And we have also clarified that our model is not meant to improve the simulations of commonly observed bulk properties, but to serve as a tool to better study phytoplankton diversity and productivity in a more realistic, turbulent ocean (*see* line 629-635).

While we would also love to compare our model outputs with Eulerian models, we realize that it is almost impossible to develop an Eulerian model that matches the features present in our IBM model. This is mainly because for an Eulerian model that allows three traits to mutate, at least $20^3$ = 8000 phytoplankton tracers (species) need to be modelled (Beckmann et al. 2019), which is computationally impossible for now (line 54-58).

The above discussion also implies an advantage of our individual-based approach in simulating phytoplankton diversity compared to the Eulerian approach. That is, our PIBM allows inferior phytoplankton ecotypes to die, thereby preserving most phytoplankton diversity without incurring additional computational cost. In comparison, we have to accommodate all phytoplankton ecotypes in the Eulerian model to simulate phytoplankton evolution, which is computationally demanding for highly diverse communities.

Despite the aforementioned difficulties in developing three-trait phytoplankton Eulerian diversity models with evolution, we have made efforts to develop simple phytoplankton Eulerian models and size-based plankton models and compared them with Lagrangian counterparts (Section 2.13.2 and 3.10).

Reference:
Beckmann, A., Schaum, C.-E. & Hense, I. 2019. Phytoplankton adaptation in ecosystem models. Journal of Theoretical Biology. 468:60–71.

Although the new model reproduces the seasonal variation of the underlying observed variables to some extent (Fig. 4), I do not expect that the results are an improvement over those from much simpler plankton models. The present model has an advantage of providing information that was not available before, but it is difficult to assess how reliable these results are (e.g., Fig. 3, 8, and 10). If validating a model directly is difficult, sensitivity tests can help.

[Response] While we admit that the results from our model may not be an improvement over those from simpler plankton models, we have to clarify that it was never our intention to provide a more accurate output of commonly measured bulk variables such as nutrient and total chlorophyll. It is well known that more complicated models tend to provide worse outputs of these variables than much simpler models (Kwiatkowski et al. 2014). However, this does not mean those efforts of developing the complicated models were worthless. The same applies to this work. The main use of our model is to provide a tool that other researchers can use to address their research questions related to phytoplankton biodiversity. These discussions have been added into our manuscript (line 629-635).

More specifically, our aim is to develop an individual-based model incorporating multiple phytoplankton traits that can facilitate our future investigations on interesting questions like what factors regulate phytoplankton diversity and how phytoplankton diversity may affect productivity. In fact, this was the main goal of our Leverhulme project. The reason why we compared the model outputs against observational data is to avoid developing a model that is far from reality. While we could, for example, test the model in an idealized setting without

having to compare it to real data, we still think it would be best that our model is capable of reproducing the major patterns of field observations. But again, we do not expect that, at the current stage, our model can outperform simpler models in terms of matching with observational data.  We have clarified this point in line 629-635. Obviously, further work should be done to validate if the model outputs are consistent with the diversity patterns observed in nature.

It will be our next step to further improve the model fit against the data which will require substantial amounts of effort and is beyond the scope of this paper.

We agree that sensitivity analyses would help model validation and enhance our understanding of the model. We have run sensitivity analyses on the number of phytoplankton particles on model outputs (Section 2.13.2 & 3.9) and how PIBM outputs differ from those of simple NPZD models (Section 2.13.2 and 3.10).

References:

Kwiatkowski, L., Yool, A., Allen, J.I., Anderson, T.R., Barciela, R., Buitenhuis, E.T., Butenschön, M. et al. 2014. iMarNet: an ocean biogeochemistry model intercomparison project within a common physical ocean modelling framework. Biogeosciences. 11:7291–304.

As many previous papers have predicted that the effect of phytoplankton acclimation is not important under the realistic ocean conditions, it is important to at least compare the results in which the effects of three phytoplankton traits (size, optimal temperature, and light affinities) are neglected. It is also interesting to see if the results without these effects differ from the Eulerian model results.

[Response] Thanks for the comment, but would you mind being more specific as to what papers showed that the effect of phytoplankton acclimation is unimportant? Please note that the definition of "acclimation" is different from "diversity" in our model. We can only have one phytoplankton species in the model (i.e., no diversity), but that species can still have

significant acclimation capacity. To our knowledge, few papers claim that acclimation is unimportant (e.g., Fennel and Boss 2003; Pahlow et al. 2013).

More controversies exist as to whether phytoplankton diversity matters, which relates to the widely studied topic of Biodiversity and Ecosystem Functioning (Vallina et al. 2017; Chen et al. 2019). While this is the ultimate goal of this study, we have limited the scope of this manuscript to describing the model and refrained ourselves discussing BEF in the paper.

To address your comment of comparing the results of the models without considering the three traits, we have developed a phytoplankton individual-based model without the three traits and its Eulerian version (see above). While we find that the patterns of the simple models are largely consistent with those of the three-trait models and the Eulerian model, we have to note that comparisons between the models depend on the parameters chosen as the simple model and the three-trait model have different parameter sets .

References

Fennel, K., Boss, E., 2003. Subsurface maxima of phytoplankton and chlorophyll: Steady-state solutions from a simple model. Limnology and Oceanography 48, 1521–1534. https://doi.org/10.4319/lo.2003.48.4.1521
Pahlow, M., Dietze, H., Oschlies, A., 2013. Optimality-based model of phytoplankton growth and diazotrophy. Marine Ecology Progress Series 489, 1–16. https://doi.org/10.3354/meps10449
Vallina, S.M., Cermeno, P., Dutkiewicz, S., Loreau, M., Montoya, J.M., 2017. Phytoplankton functional diversity increases ecosystem productivity and stability. Ecological Modelling 361, 184–196. https://doi.org/10.1016/j.ecolmodel.2017.06.020
Chen, B., Smith, S.L., Wirtz, K.W., 2019. Effect of phytoplankton size diversity on primary productivity in the North Pacific: trait distributions under environmental variability. Ecology Letters 22, 56–66. https://doi.org/10.1111/ele.13167

The model description is unclear, making it difficult to understand how the model is calculated.
I could not find the procedure for calculating the three traits in the paper. The proper explanation how these traits for Lagrangian plankton particles are calculated, except

mutation (L194), and how their initial conditions are given, especially temperature and light affinity. I cannot understand the relationship between Cdiv, ESD, and V (in Table 2), while all these variables appear to represent the cell size. Nor can I understand how these traits are affected by the motion of Lagrangian particles such as zi. Note that this is the most essential part of the new model.

[Response] We appreciate the reviewer's feedback and have carefully revised the model description to improve clarity.
In lines 107-109, we now explicitly clarify how the three traits that characterize each super-individual are defined. Additionally, in the "Initial Conditions" section, we provide details on the ranges of values used in our simulations. At the start of each simulation, each individual is assigned values for these three traits, which remain constant throughout the simulation, except for the possibility of mutation during duplication events.
The legend of Table 2 was revised to better clarify how allometric relationships are applied in this study.

It is important to mention that the inclusion of more empirical formulae does not necessarily improve the model. The authors tried to include a large number of existing empirical formulae without justifying why they should be included. The phytoplankton model largely follows that by Geider et al. (1998). Here it is important to specify how the new model differs from Geider et al. (1998). It is not clear whether the zooplankton model used the existing model, or developed a new model. Again, the model equations are listed without any explanation to justify their use.

[ Response] Again, we have carefully revised the text, ensuring that our modifications to the base  model of Geider et al. (1998) are well-justified. Our changes focus on incorporating photoinhibition, which is essential for capturing the decline in photosynthetic efficiency under saturating irradiance. This ensures that simulated carbon fixation remains within physiologically realistic limits (Falkowski and Raven, 2007). Additionally, we now account for the effect of temperature on the maximal carbon-specific photosynthesis rate by replacing the original Geider et al. (1998) term $P^{C}_{ref}$, maximal photosynthetic capacity at a reference temperature, with $\mu_m$. This modification explicitly incorporates temperature dependence, following the analyses of Chen et al. (2022) which is well founded on empirical data analyses and the theory of Metabolic Theory of Ecology. The size scaling relationships largely follow

the excellent study by Ward et al. (2012). While we agree that these are largely empirical relationships, but we think this is the best we can use right now.

As a result, our approach provides a more dynamic and accurate representation of phytoplankton growth rates compared to a static reference value.

We agree that the equation describing the tradeoff between $\alpha^{Chl}$ and photo-repair rate is less well studied. We have discussed this in Section 4.1.

We also clarified the description of the zooplankton model, emphasizing that the model was following Ward et al. (2012) (Section 2.3.2).

References:
Chen, B. 2022. Thermal diversity affects community responses to warming. Ecological Modelling. 464:109846.
Falkowski, P.G. and Raven, J.A., 2013. Aquatic photosynthesis. Princeton University Press.
Geider, R.J., MacIntyre, H.L. & Kana, T.M. 1998. A dynamic regulatory model of phytoplanktonic acclimation to light, nutrients, and temperature. Limnology and Oceanography. 43:679–94.

Other points:
L.205: R package TPD - necessary to explain

[Response] Due to a problem we found in the TPD R package, we have changed the method of calculating functional diversity from TPD to the Rao index which has been explained in Section 2.11.

L305: no explanation on how to calculate the size spectra of phytoplankton

[Response] This section was also improved, giving more details on calculating the size spectra for both fractions, phyto- and zooplankton (Section 2.12).

L325: Information on the temporal resolution of forcing and on the type of the 1D model is

necessary.

[Response] Details included in Section 2.4.

If the 1D model includes nonlocal mixing, Kv does not necessarily predict the vertical motion of particles.

[Response] Apologies, we forgot to update the text in the manuscript. Initially the Kv profiles were generated using the kpp scheme, but later we used the Kv profiles shared by Dr. Sergio Vallina which were computed using the k~ε turbulence model in the 1D GOTM model (Burchard et al. 1999; Bruggeman and Bolding 2014). Now the text has been correctly updated (Section 2.4; line 335).

**References:**
Bruggeman, J. & Bolding, K. 2014. A general framework for aquatic biogeochemical models. Environmental Modelling & Software. 61:249–65.
Burchard, H., Bolding, K., Ruiz Villarreal, M., 1999. GOTM e a General Ocean Turbulence Model. Theory, Applications and Test Cases. Tech. Rep. EUR 18745 EN. European Commission.

L346: I should mention that the calculation of zi using random walk can exaggerate the variance of traits, because it neglects the correlation of two particles located nearby. It also neglects the effects of horizontal diffusion and patchiness.

[Response] Thanks for the comment. First, we admit that we did neglect the effects of horizontal diffusion since we are using a one-dimensional model. If we couple our Lagrangian model with a two- or three-dimensional model, horizontal diffusion will of course be considered.

However, we think that patchiness has already been implicitly considered for phytoplankton which is modelled as particles.

And we are not fully convinced that calculating Zi using random walk overestimates the variance of phytoplankton traits. We are following the classical study of Visser (1997) who

derived the vertical motions of particles analytically (see Appendix 1 in his paper ). If you have better recommendations, please let us know and we will be grateful.

Reference:
Visser, A.W., 1997. Using random walk models to simulate the vertical distribution of particles in a turbulent water column. Marine Ecology Progress Series 158, 275–281. https://doi.org/10.3354/meps158275

L371: The Neumann boundary condition does not mean no flux at the boundary.

[Response] Yes, thanks for pointing it out. Neumann boundary condition simply means constant flux at the boundary. We have corrected the text to "zero fluxes are applied…".

Fig. 2: It is unrealistic that MLD is zero during the summer.

[ Response] Thank you for pointing it out. We found a mistake in our code and have resolved it.

Fig. 11: error in the caption (red solid lines for summer).
[Response] Corrected, thank you.

L547: This section should be moved to the introduction. It may also be important to include references of Lagrangian plankton models using different approaches such as Jokulsdottir & Archer (2016), Kida & Ito (2017), and Noh et al. (2021). It is especially so, when references of Lagrangian cloud models are included.

[Response] As suggested, we have shortened this section and moved it to the Introduction (see lines 38-49). We also added the three references you suggested to the text.

Review 2

In general, individual-based phytoplankton model is a promising topic. The authors built a vertical-1D NPZD model at BATS, in which only "P" is individual-based. Though building a model from scratch involves a lot of work (and I value the efforts the authors put), sometimes it's actually not necessary cause there're already models available to use to answer research questions the authors wanted to ask (and probably better in performance and easier to use). For example, models developed by Hellweger's group (e.g., Hellweger et al. 2014, Science, mutation is included) and PlanktonIndividuals.jl (https://github.com/JuliaOcean/PlanktonIndividuals.jl, developed by MIT Darwin Group, the model resolves intracellular macromolecular composition of multiple species). Personally, I think collaboration with these groups is actually more efficient than building a model from scratch. Anyway, there's nothing wrong the authors chose to build their own model. But I still have some question and comments for the authors before moving forward to publication.

[Response] We appreciate the reviewer's recognition of the work involved in building a new model. While we understand that there are existing models such as those from Hellweger's group and PlanktonIndividuals.jl, we believe that these models are insufficient to fully address the research questions we aim to explore. For example, our model is designed to incorporate the individual-based dynamics of phytoplankton, with an emphasis on traits such as size, temperature preference, and light affinity, which are not fully captured by other models. Both Hellweger's group and PlanktonIndividuals.jl focus more on the phytoplankton intracellular physiology and biogeochemistry (Hellweger 2020; Wu and Forget 2022). In contrast, our model focuses on addressing the question of what regulates phytoplankton functional diversity in the ocean and what is the effect of phytoplankton diversity on primary productivity. The food web structure in our model is also much more complicated than Hellweger's models and PlanktonIndividuals.jl.

Another obvious reason for our independent model development is that we started to develop the model much earlier (around 2019) than when we were aware of PlanktonIndividuals.jl.

Of course, we totally agree that collaboration with these groups is definitely a valuable way going forward. In fact, one of us (B. Chen) has spoken to Dr. Zhen Wu, the author of PlanktonIndividuals.jl, in person in January 2025 and discussed potential collaborations in the

future. During the discussion, we also learned more about the PlanktonIndividuals.jl model. Ultimately, we might convert our fortran codes into the Julia code and merge our model with PlanktonIndividuals.jl (line 649); but this will take a few years to complete.

References
Hellweger, F.L. 2020. Combining Molecular Observations and Microbial Ecosystem Modeling: A Practical Guide. Annual Review of Marine Science. 12:267–89.

Wu, Z. and Forget, G., 2022. PlanktonIndividuals. jl: a GPU supported individual-based phytoplankton life cycle model. Journal of Open Source Software, 7(73), p.4207.

**Specific comments**

- Line 20: This is true, but only 20000 particles still cannot represent the full distribution of phytoplankton community. It's too low compared to the real cell density.

[Response]  Agree, but we have to reach a balance between computation speed and the representation of phytoplankton trait distribution.

While it is true that 20,000 particles cannot explicitly represent all phytoplankton cells present in the region, no model—whether Eulerian or Lagrangian—can fully resolve the entire phytoplankton community. Eulerian models also rely on predefined functional groups, limiting species diversity representation. Our comparison between different particle numbers (see Figure 12) demonstrates that phytoplankton biomass, production, and diversity remain stable, suggesting that our choice of 20,000 particles is sufficient to capture the dominant ecological dynamics while remaining computationally efficient.

- Line 23: Lagrangian phytoplankton model may also not be the ground truth.

[Response] We have modified the text to "Uncertainty remains about whether Eulerian models or fixed-depth in situ incubations overestimate or underestimate primary production compared to the realistic situation in which phytoplankton cells can move in the water column".

- Line 45: This is not true. PlanktonIndividuals.jl as mentioned above, also developed by Mick Follows' group.

[Response] As explained above, PlanktonIndividuals.jl incorporates limited phytoplankton diversity.

- Line 74: a constant number of super-individuals with each super-individual represents a flexible number of cells VS flexible number of super-individuals with each super-individual represents a constant number of cells? Actually I prefer the second one. Here's the reason, the first one sounds like an assemble of many tiny 0D Eulerian models and cell division is actually a change of concentration not number of cells.

[Response] We acknowledge the reviewer's perspective regarding the representation of super-individuals. However, we chose a fixed number of super-individuals with a flexible number of cells per super-individual for computational efficiency and practical model implementation. This approach has been successfully applied in previous studies (Sheffer et al. 1995 and others) and allows for an ecologically meaningful representation of phytoplankton dynamics, including the influence of zooplankton grazing on abundance.

While the alternative approach (a flexible number of super-individuals) could explicitly track cell numbers, it introduces significant computational complexity without necessarily improving the realism of macroscopic population dynamics. The main issue is the memory issue. If we keep the number of cells per super-individual constant, the number of super-individuals may vary by several orders of magnitude during simulation which can sometimes blow up the simulation.

Reference:
Scheffer, M., Baveco, J.M., DeAngelis, D.L., Rose, K.A., van Nes, E.H., 1995. Super-individuals a simple solution for modelling large populations on an individual basis. Ecological Modelling 80, 161–170. https://doi.org/10.1016/0304-3800(94)00055-M

- Eq.3: I don't think respiration cost Chla.

[Response] Yes, respiration does not necessarily "cost" chlorophyll content in the sense of direct degradation or loss. However, the inclusion of the respiration term accounts for the energy cost of maintaining phytoplankton metabolism, which is essential for balancing chlorophyll production and degradation under variable environmental conditions. This term has been included in a number of impactful studies (Geider et al. 1998; Pahlow et al. 2013).

References:

Geider, R., Hl, M., Tm, K., 1997. Dynamic model of phytoplankton growth and acclimation: responses of the balanced growth rate and the chlorophyll a:carbon ratio to light, nutrient-limitation and temperature. Marine Ecology Progress Series 148, 187–200. https://doi.org/10.3354/meps148187

Pahlow, M., Dietze, H., Oschlies, A., 2013. Optimality-based model of phytoplankton growth and diazotrophy. Marine Ecology Progress Series 489, 1–16. https://doi.org/10.3354/meps10449

- Fig.4: There're still considerable discrepancies between simulations and observations. For example, the deep chl maximum and the depth of NPP. I'll put a question mark on the choices of parameters.

[Response] We agree there are still discrepancies between PIBM output and data. But this is the best we can achieve now given the complexity of the model and the challenges in parameter tuning. We have stressed that it is never our intention to provide an accurate output of commonly measured bulk variables such as nutrient and total chlorophyll. The main use of our model is to provide a tool that can be used to study phytoplankton biodiversity. These discussions have been added to our manuscript (before Section 4.1). We have discussed the possible reasons for the discrepancy in Section 4.1.

- Line 535: As mentioned above, 20000 particles may still not be enough.

[Response] We have already responded to this concern.

- Section 4.1 and 4.2 should be shortened and go to Introduction.

[Response] We have shortened Section 4.1 and moved it to the Introduction. However, we think it would be better to keep Section 4.2, as it highlights the strengths and limitations of our model.

We hope the above responses satisfactorily address the reviewers' comments. Thanks again for your efforts and insightful comments that have helped improve the manuscript.

Best regards,
Iria Sala & Bingzhang Chen

---

## Author Response (AR2)

**Dear Editor,**

We are grateful to the reviewers and to you for your constructive feedback. Below, we provide detailed responses (in red) to each of the comments raised.

**Reviewer 1**

My first feeling after reading the whole Intro is that the authors wanted to build a model that can overcome the limitations mentioned in paragraphs from line 22 to line 53. But to prove that, the new model must have better performance than previous models. However, the model performance is actually worse than simple Eulerian models. This is totally okay, but the Intro should be modified to better express the authors true focus, that is to serve as a tool to better study phytoplankton diversity and productivity in a more realistic turbulent ocean.

[Response] We thank the reviewer for this helpful comment. We have revised the Introduction to clarify that the objective of our study is not to improve the performance of phytoplankton models in reproducing bulk properties, but rather to provide a new tool to explore phytoplankton diversity, acclimation, and productivity in a dynamically variable environment.

**## Specific Comments**

Line 24: This is not true. To my understanding, the change of phytoplankton concentration in Eulerian models actually means the cells are moving.

[Response] We have revised the text to better explain the distinction between the Eulerian and Lagrangian frameworks, specifically focusing on how they represent the movement and environmental experience of phytoplankton (see lines 46-47).

Line 59: increasingly what? popular?

**[Response] Yes, popular. Thank you.**

Eq3. I still don't think respiration cost Chla. Actually, I read Geider et al. 1997 again to double check if I'm wrong. Geider used a Eulerian framework, so the degradation of Chla is actually caused by mortality, but Chla doesn't degrade within the cell. That's the difference.

[Response] That is correct, Geider et al. (1997) present a population-level model in which chlorophyll loss is attributed to cell-specific mortality. However, our formulation follows the intracellular framework developed in Geider et al. (1998), where their Eq. 3 includes a firstorder degradation term for intracellular Chl-a. This term represents a physiological pigment loss process occurring within the cell, and is not associated with mortality or population turnover.

Nevertheless, the reviewer is right in highlighting the fact that  $R_C$ ,  $R_N$ , and  $R_{Chl}$  are degradation rate constants, not respiration rates. We thank the reviewer for drawing attention to this and have clarified the terminology in the manuscript (see lines 125-128).

Line 656: DCM is actually not captured. The patterns are quite different (Fig.4, middle panels). DCM is better simulated in the other two models (Fig.13). It would be interesting if the authors can explain why this happens. This is actually more meaningful.

[Response] Thank you for the note. The original explanation didn't describe the model differences clearly enough. We've now expanded the text to explain this variability in more detail, focusing on the effects of trait selection, acclimation, photoinhibition, and self-shading in PIBM. The main reason for this to occur is because trait selection favour types with elevated growth rates at higher light levels.

We hope the above responses and revisions are satisfactory. Thanks again for your input.

Best regards, Iria Sala, Bingzhang Chen

---

## Author Response (AR3)

Dear Editor,

Thank you so much for your helpful comments. They have helped us pay attention to every detail and improve the manuscript. We spotted and corrected several areas of unclear or missing details and some errors. These revisions have strengthened the overall quality of the manuscript.

- Figures 2, 4, 6, 8 and 13 are composed of raster graphics, please use vector graphics as in all other plots.
[Response] We have changed all these plots to vector graphics.

- Avoid using italics for subscripts and superscripts which do not denote variables: "C", "N", Chl", "div", "opt", "phy", "dn", "ref", "max", "min", "m" (?), "PSII", etc.
[Response] Done.

- Avoid using italics for units (and use LaTeX \upmu instead of $\mu$).
[Response] Done, thank you.

- The Github and Zenodo links lead to code version labelled "second release", while the title mentions PIBM 1.0 (which is an older release not accessible via Zenodo) - please update the paper title to match the relevant code release
[Response] We have made a new release which is now published on Zenodo and matched with the paper title (10.5281/zenodo.15296286).

- It is somewhat misleading that the software name PIBM is almost nowhere to be found in the code archive (where IBM is used) - please update the code archive with a project name and version matching the paper.
[Response] We have updated the Readme file of the code archive to match the paper description.

- Spell openMPI as "Open MPI" (2 instances: p34 & p15).
[Response] Changed!

- Note: OpenMPI is just a particular implementation of the MPI standard, statements such as "use OpenMPI parallel computing" sound misleading - it is MPI API that the code relies on, and it can be compiled against different MPI implementations, Open MPI being just one example.
[Response]  Thank you for your comment. You are absolutely right, and we have updated the text for clarity. The revised lines in the manuscript are as follows:

- Line 381: "Due to the computational intensity of the particle random walk, we implemented parallel computing using the MPI standard,
- which can be compiled with different implementations such as Open MPI (Message Passing Interface Forum, 2023)."
- Line 658: "We also implemented parallel computing using MPI to simulate the random walk of both passive and phytoplankton particles, with Open MPI as one implementation option."

- I see that the Gabriel et al. 2004 citation is suggested on the Open MPI website as a general-purpose citation for Open MPI, but it is a 20-year old conference paper about a draft of one particular MPI-2 implementation, while we are now at MPI-4.1!
[Response] As shown above, we now refer to the technical report by the Message Passing Interface Forum (2023) for the MPI standard.

- Spell openMP as OpenMP (p34/l660).
[Response] Done.

- Since you mention considering integration with PlanktonIndividuals.jl and point to the Julia language, please clarify in the paper that the IBM codebase is FORTAN/Matlab/shell/R.
[Response] Detail added to the Model description: Overview section (Line 82).

- Fix URL in Ledbetter 1979 reference (htps://doi.org/https://doi.org/...)
[Response] Corrected.

Best regards,
Iria Sala
Bingzhang Chen